

# An Adaptive Semi-Lagrangian Advection Model for Transport of Volcanic Emissions in the Atmosphere

Elena Gerwing[1], Matthias Hort[1], Jörn Behrens[2], and Bärbel Langmann[1]

[1]Institute of Geophysics, University Hamburg, Bundesstr. 55, 20146 Hamburg, Germany
[2]Department of Mathematics, Differential Equations and Dynamical Systems, Bundesstr. 55, 20146 Hamburg, Germany

*Correspondence to:* Elena Gerwing (elena.gerwing@uni-hamburg.de)

**Abstract.** Dispersion of volcanic emissions in the Earth atmosphere is of interest for climate research, air traffic control as well as human wellbeing. Current volcanic emission dispersion models rely on fixed grid structures that often are not able to resolve the fine filamented structure of volcanic emissions while being transported in the atmosphere. Here we extend an existing adaptive semi-Lagrangian advection model for volcanic emissions including the sedimentation of volcanic ash. The advection of volcanic emissions is driven by a pre-calculated wind field. For evaluation of the model, the explosive eruption of Mount Pinatubo in June 1991 is chosen, which was one of the largest eruptions in the 20th Century. We compare our simulations of the climactic eruption on June 15, 1991 to satellite data of the Pinatubo ash cloud and evaluate different sets of input parameters. We could reproduce the general advection of the Pinatubo ash cloud and owing to the adaptive mesh, simulations could be performed at a high local resolution while minimizing computational cost. Differences to the observed ash cloud are attributed to uncertainties in the input parameters and the pass by of Typhoon Yunya, which is probably not completely resolved in the wind data used to drive the model. Best results were achieved for simulations with multiple ash particle sizes.

## 1 Introduction

Tephra and $SO_2$ emissions from large volcanic eruptions have a crucial impact on short- and long-term climate variations, air traffic and the living conditions of people in the surrounding of volcanoes. Large tropical and high-latitude eruptions were primary drivers of interannual-to-decadal temperature changes in the Northern Hemisphere during the last 2500 years (e.g., Sigl et al., 2015). However, even smaller volcanic eruptions do significantly affect the living conditions on a local scale. For example, the respiration of volcanic ash and gas (e.g., Horwell and Baxter, 2006) is along with the fall of tephra (e.g., Paladio-Melosantos et al., 1996) the most important impact on the local scale. Heavy tephra falls can lead to collapse of buildings, destruction of mechanical and electrical systems, disruption of transport systems, formation of enormous lahars, chemical and physical changes in water quality and damage of vegetation, crops, forestry and pastures (Folch, 2012). Drifting ash clouds pose a serious thread to jet aircraft and can lead to engine failure (e.g., Casadevall, 1993). Since 1976 an average number of two damaging encounters per year between aircraft and ash clouds has been reported (Guffanti et al., 2010), and Clarkson et al. (2016) lately reviewed available engine and volcanological data and proposed a new 'Safe-to-Fly' chart with a much lower ash concentration threshold than previously recommended.



Volcanic $SO_2$ injected into the stratosphere has a global impact by its conversion to sulphate aerosol which disturbs the Earth's radiation balance. Tropical volcanic eruptions thereby lead to warmer winters and colder summers on the Northern Hemisphere continents (Robock, 2000). In addition, volcanic aerosols lead to an increase in stratospheric particle surface area, enhancing the ozone destruction especially in high latitudes (Solomon, 1999). The amplitude of the diurnal cycle of the surface

air temperature is reduced by volcanic tephra remaining in the atmosphere on timescales from minutes to weeks (Robock, 2000).

In order to mitigate risks and assess hazards originating from volcanic clouds, accurate observations and forecasts are needed. Advecting volcanic clouds can be tracked by satellite observations, but satellite images in the visible spectrum only result in outer contours of the cloud. Moreover, the global coverage and image frequency of satellite observations is highly

inhomogeneous and satellite images only reflect the current state and can not be used for forecasting. Therefore, numerical models predicting the advection of ash or $SO_2$ are necessary.

There are several models simulating the advection (and sedimentation) of ash and $SO_2$ clouds. They are mainly performed on a regular grid and can generally be divided into two types by their numerical framework: *Eulerian models* like ATHAM (Oberhuber et al., 1998), REMOTE (Langmann, 2000), Fall3d (Folch et al., 2009) or Ash3d (Schwaiger et al., 2012) and

*Lagrangian models* including Puff (Searcy et al., 1998) and NAME III (Jones et al., 2007). Additionally, there are some models using other approaches like semi-analytical tephra transport and dispersion models (HAZMAP (Macedonio et al., 2005; Pfeiffer et al., 2005) or TEPHRA (Bonadonna et al., 2005)) and the Lagrangian-Eularian model Vol-CALPUFF (Barsotti et al., 2008) and a volcanological adaptation of HySplit (Stein et al., 2015). For more details on these models the reader is referred to a recent review by Folch (2012).

In this article, we extend an existing semi-Lagrangian advection model performed on an adaptive, triangulated mesh (*Amatos* and *Flash*: Behrens (1996); Behrens et al. (2000)) for volcanic emissions. The semi-Lagrangian method has the advantage of a very stable and numerically efficient advection calculation and can be performed in parallel (Behrens, 1996). Adaptive mesh methods have the additional advantage of high resolution in the area where the advected cloud currently resides, while the computational cost is kept relatively low by using a coarse mesh ouside the cloud. With this model, simulations forecasting

the advection of an ash cloud for several days could be performed at very low computational cost (CPU times of seconds to minutes). We apply our new model to the advection and sedimentation of tephra, because $SO_2$ clouds cover a much larger area and the sedimentation of tephra occurs in timescales of minutes to weeks, while $SO_2$ and sulphate can remain in the atmosphere for some years. We concentrate on the advection of the ash cloud, neglecting complex eruption column dynamics and the influence of the eruption column on the surrounding atmosphere. This is a valid assumption, because far enough from

the vent, these effects play a minor role (Folch, 2012).

In the following we first introduce the implementation of the adaptive semi-Lagrangian advection algorithm and explain how the sedimentation of particles has been implemented into this model. We then turn to the description of our case study of the climactic eruption of Mt. Pinatubo, 1991. Here we focus first on the main advantages of our solution and then carry out a sensitivity analysis by varying different input parameters. We finish with some discussion (including a detailed performance

study) and conclusion.





## 2 Model description

In the model we solve the advection equation

$$\frac{\mathrm{d}C}{\mathrm{d}t} = R, \text{ with } \frac{\mathrm{d}C}{\mathrm{d}t} = \frac{\partial C}{\partial t} + \boldsymbol{u} \cdot \boldsymbol{\nabla} C. \tag{1}$$

$R$ is the so called right hand side which can include additional forces as well as sources and sinks for the scalar tracer concen-
tration $C(x, y, z)$. In our case $R$ includes the volcanic source (for implementation in our case see below).

The advection of the cloud is driven by a pre-calculated wind field $\boldsymbol{u}(x, y, z)$ from the regional scale atmospheric chemistry
and climate model REMOTE (**Re**gional **Mo**del with **T**racer **E**xtension), for details see (Langmann, 2000). Initial meteorological
data are taken from the ECMWF and boundary conditions are updated every 6 h. The horizontal resolution of the wind field
is $0.5° \times 0.5°$ (approximately 55 km × 55 km). In the vertical direction a pressure-sigma coordinate subdivides the model
atmosphere into 31 layers of increasing thickness between the Earth surface and the 10 hPa pressure level. Here we utilized all
vertical layers and interpolated the wind in $x-$, $y-$ and $z-$direction to the grid resolution.

We solve the time dependent part of this equation by a semi-Lagrangian method (see (Staniforth and Cote, 1990)). A
finite element like method was used to solve for the spatial dependence of $C$. The solution is carried out on an adaptive
mesh: in regions, where a high spatial resolution is required, the mesh is refined, whereas the mesh size in other parts of
the model is kept relatively coarse. Thereby, memory requirements can possibly be decreased by orders of magnitude with-
out losing accuracy (Behrens, 1996). In our case the refinement criterion is based on the concentration gradient $\boldsymbol{\nabla} C|_{\tau_i}$ in a
mesh element $\tau_i \in T$, where $T$ represents the complete triangulation. A mesh element is refined if $\boldsymbol{\nabla} C|_{\tau_i} > \theta_{\mathrm{ref}} \cdot \boldsymbol{\nabla}_{\max}$, with
$\boldsymbol{\nabla}_{\max} = \max_{\tau_i \in T}\{\boldsymbol{\nabla} C|_{\tau_i}\}$ being the maximum of all local concentration gradients. Accordingly, a mesh element is coars-
ened if $\boldsymbol{\nabla} C|_{\tau_i} < \theta_{\mathrm{crs}} \cdot \boldsymbol{\nabla}_{\max}$. The parameters $\theta_{\mathrm{ref}}$ and $\theta_{\mathrm{crs}}$ (with $0 < \theta_{\mathrm{ref}} \leq 1$ and $\theta_{\mathrm{crs}} < \theta_{\mathrm{ref}}$) define the relative tolerances for
refinement and coarsening respectively.

### 2.1 Particle Sedimentation

The sedimentation of tephra from an advecting ash cloud is mainly dependent on the grain size, the density of the particles
and the properties (viscosity and density) of the surrounding air. In order to account for sedimentation, the terminal settling
velocity $v_t$ (balance between drag force and gravitational force) is calculated for atmospheric conditions at every mesh point
and every time step. The terminal settling velocity is given by

$$v_t = \sqrt{\frac{4}{3}\frac{(\rho_p - \rho)}{\rho C_d}D_p g}, \tag{2}$$

with $\rho_p$ the density of an ash particle with a diameter $D_p$, $\rho$ the density of the surrounding fluid (calculated here from REMOTE
simulation results), $g$ the gravitational acceleration and $C_d$ the drag coefficient. Settling of particles is then accounted for in the
advection equation (Eq. 1)

$$\frac{\partial C}{\partial t} + u_x \frac{\partial C}{\partial x} + u_y \frac{\partial C}{\partial y} + (u_z - v_t)\frac{\partial C}{\partial z} = R, \tag{3}$$





by modifying the vertical advection term.

The terminal settling velocity is dependent on the drag coefficient $C_d$ which is a function of the Reynolds number Re which in turn depends on the settling velocity. Empirical formulations of the drag coefficient for different regimes of the Reynolds number have been suggested by several authors (Seinfeld and Pandis, 2006; Dellino et al., 2005; Bonadonna et al., 1998;

Herzog et al., 1998; Ganser, 1993; Arastoopour et al., 1982; Wilson and Huang, 1979). Here we use the model introduced by Ganser (1993), which gave the best results for our conditions (look at the online supplement for more details).

For calculating Re the dynamic viscosity of the air in dependence on the air temperature is required (Pruppacher and Klett, 1997):

$$
\mu =
\begin{cases}
(1.718 + 0.0049 \cdot T_C) \times 10^{-5} & T_C \geq 0^{\circ}\text{C} \\
(1.718 + 0.0049 \cdot T_C - 1.2 \times 10^{-5} \cdot T_C^2 \times 10^{-5} & T_C < 0^{\circ}\text{C},
\end{cases}
\tag{4}
$$

with $T_C$ the temperature in $^{\circ}$C.

Particles require a certain time (a so called *relaxation time*) to reach their terminal settling velocity (Seinfeld and Pandis, 2006):

$$
t_r = \frac{M_p C_s}{3 \pi \mu D_p}.
\tag{5}
$$

Here, $M_p$ is the mass of the ash particle and $C_s$ is a slip correction factor defined by Seinfeld and Pandis (2006).

Using reasonable values for the parameters in (5), it is obvious that the maximum time required by particles to reach their terminal settling velocity is relatively short. Even for particles with a diameter of 1 mm ($\phi = 0$) the maximum relaxation time is only about 9 s. Compared to the default simulation time step of 10 min used here and a simulation period of about five days, the relaxation time is negligible. Therefore, we assume that the ash particles are falling directly with their terminal settling velocity.

With the assumption that in dilute clouds ash particles of different sizes do not affect each other but each particle settles individually (i.e. we neglect particle aggregation as well as particle particle interaction) we apply the model for individual particle diameters and then combine the results of different runs to predict the sedimentation of the complete grain size distribution. As we model the fallout from the umbrella cloud, we furthermore assume that the ash particles already reached their maximum injection height at the start of the model. Complex eruption column dynamics are neglected and we suppose no interaction

with and no re-entrainment into the eruption column. In addition, the settling of ash particles is strongly affected by rain fall and particle aggregation (see e.g. Brown et al. (2012)). Since this work is a first case study of the modeling of sedimentation of ash particles on an adaptive mesh the impact of rain on the sedimentation and aggregation of ash particles is neglected. Finally, we do not monitor the thickness of the ash deposited on the ground.





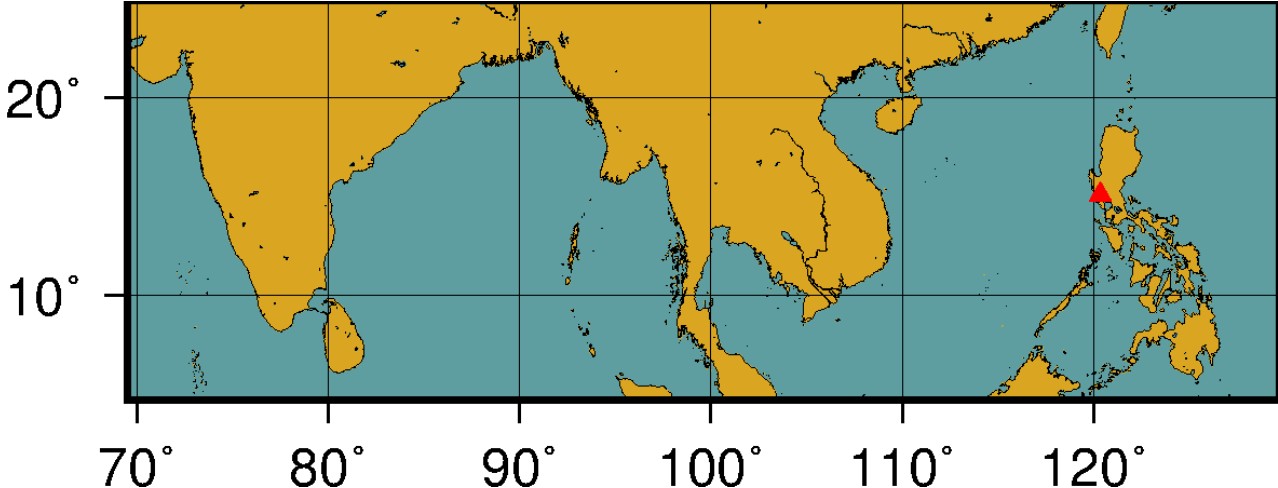

**Figure 1.** Location of Mt. Pinatubo marked by the triangle in the simulation domain (69.5°E/4.5°N to 130°E/25.5°N).

## 3 Modeling the Climactic Eruption of Mt. Pinatubo

### 3.1 Summary of the 1991 Mt. Pinatubo

The 1991 Mt. Pinatubo eruption on the island Luzon in the Philippines (see Fig. 1) was one of the largest explosive eruptions in the 20th Century. The amount of erupted $SO_2$ induced a global cooling of at least 0.5°C in the two years following the eruption

(Self et al., 1996). Pyroclastic flows, lahars (thick volcanic mudflows) and ash fall made more than 50,000 people homeless, affected the lives of more than a million people and caused 200 to 300 deaths (Punongbayan et al., 1996; Bautista, 1996).

Following a quiet period of about 500 years (Newhall et al., 1996) activity at Mt. Pinatubo started in July 1990 with a magnitude 7.8 earthquake along the Philippine fault, about 100 km north-east of the summit (Punongbayan et al., 1991). Many smaller earthquakes were recorded in the following months and in April 1991, the first eruptions with column heights between

1 and 8 km took place. The explosive phase began on June 12, 1991 and lasted till June 16 with subplinian to plinian eruptions and column heights between 19 and 40 km. The most violent eruptions occurred between 13:40 (PDT) and 22:40 on June 15 with more or less continuous high-output activity. The intensity of this eruption period began to decrease after about three hours at 16:40 on June 15 (Wolfe and Hoblitt, 1996). In the nine hour lasting climactic eruption phase, 80 percent of the total erupted volume was ejected and the highest eruption columns were reached (Holasek et al., 1996).

During the first phase of the climactic eruption, the ash expanded radially and formed a huge umbrella cloud. Koyaguchi and Tokuno (1993) analyzed the hourly multi-spectral images of the Global Mapping Satellite (GMS) on June 15, 1991 and showed that following the onset of the climactic eruption at 13:41, the erupted material expanded radially for about five to six hours in a giant umbrella cloud. At 14:40 Koyaguchi and Tokuno (1993) identified an ash cloud of 280 km diameter in the satellite images and at 15:40 the umbrella cloud covered an area with a diameter of 400 km. Similar studies using infrared

(Lynch and Stephens, 1996) and GMS-4 visible band data were used to determine the radial expansion and advection of the





| Eruption phase start | Eruption phase end | Mass eruption rate [kg/m$^3$] |
|---|---|---|
| June 13, 1991, 08:25 | June 13, 1991, 08:55 | $5.775 \times 10^7$ |
| June 14, 1991, 13:09 | June 15, 1991, 13:41 | $2.1 \times 10^7$ |
| *June 15, 1991, 08:10* | June 15, 1991, 10:27 | $9.75 \times 10^6$ |
| *June 15, 1991, 10:27* | June 15, 1991, 13:41 | $2.25 \times 10^7$ |
| *June 15, 1991, 13:41* | June 15, 1991, 22:41 | $2.1 \times 10^8$ |
| *June 15, 1991, 22:41* | June 16, 1991, 10:41 | $1.5 \times 10^7$ |

**Table 1.** Eruption phases implemented in the right-hand side. For time periods not listed in this Table, the right-hand side was set to zero. The values given in italics are those for the climactic phase of the eruption. For converting m$^3$/s to kg/s we used an average density of 1500 kg/m$^3$.

ash cloud in west-southwest direction. The southwestward advection of the umbrella cloud mainly reflects the wind direction in the stratosphere (Koyaguchi and Tokuno, 1993).

Light to moderate tephra was displaced southward and moderate to heavy tephra northeastward by Typhoon Yunya which passed in a distance of about 75 km northeast of the erupting volcano at around 14:00 on June 15, 1991 (Oswalt et al., 1996).

This atypical wind in the lower and middle troposphere caused the wide distribution of tephra in nearly all directions around the volcano (Wolfe and Hoblitt, 1996). The heaviest tephra falls occurred during the climactic eruption on June 15, producing tephra fall deposits with up to 33 cm thickness (Paladio-Melosantos et al., 1996). An area of around 7500 km$^2$ on Luzon was covered by more than 1 cm thick tephra deposits and the entire island obtained at least a trace of ash (Paladio-Melosantos et al., 1996). Paladio-Melosantos et al. (1996) examined the grain sizes of the Pinatubo 1991 tephra-fall deposits on the Luzon

Island relatively close to the vent ($\leq$ 30 km distance), while Wiesner et al. (1995) recorded the fallout of tephra following the climactic eruption by two sediment traps moored at 14.60°N and 115.10°at a water depth of 1190 m and 3730 m in the South China Sea.

## 3.2   Simulation set-up

### 3.2.1   Volcanic Ash Emissions

In order to properly model the source term $R$ in (3), mass eruption rates need to be defined as model inputs. Our estimate of the mass eruption rates is based on observations by Holasek et al. (1996) in visible and infrared satellite images. For some satellite data Holasek et al. (1996) could not determine the altitude of the eruption plume and we completed values with data from Self et al. (1996). A complete list of eruption height used to estimate mass eruption rates is given in Tab. A1. After June 16 10:41, secondary explosions were induced by the interaction of the hot ignimbrite with water, but these secondary eruption

plumes were of less intensity and are not considered in this study. Holasek et al. (1996) calculated an average eruption rate of $1.4 \times 10^5$ m$^3$/s for the nine hour lasting climactic phase from June 15 13:41 to 22:41. Accordingly, we estimated mass eruption rates for the other eruption phases from the data of Holasek et al. (1996) and Self et al. (1996). The mass eruption rates used in this study are listed in Tab. 1.



| Parameter | Standard value | Variation range | Units |
|---|---|---|---|
| Fine mesh level | 17 | $14 - 23$ | – |
| Coarse mesh level | 8 | – | – |
| Tolerance of refinement $\theta_{\mathrm{ref}}$ | 0.02 | – | – |
| Tolerance of coarsening $\theta_{\mathrm{crs}}$ | 0.005 | – | – |
| Time step length | 600 | – | seconds |
| Number of time steps | 684 | – | – |
| Initial cloud radius | 4 | $2 - 5$ | degree |
| Height of initial cloud center | 17 | $15 - 21$ | km |
| Initial cloud thickness | 6 | $2 - 8$ | km |
| Grain size | 4.5 | $0 - 8$ | $\phi$ |

**Table 2.** Parameter values used in the model calculations.

Ash is not released evenly along the eruption column into the atmosphere but mostly close to the neutral buoyancy level. Fero et al. (2009) simulated the Pinatubo eruption with different tephra dispersal models and determined that most of the ash was advected in an umbrella cloud at the level of the tropopause at around 17 km – significantly below the maximum column heights of 40 km and below the main transport level of $SO_2$ at around 25 km. We follow their approach and inserted ash in a

4 km long cylinder with a radius between 2 and 5 degrees and a medium height of 17 km centered above Pinatubo volcano (see also Tab. 2). The size of the cylinder is varied during the sensitivity study as well as its center location in the atmosphere. By dividing the mass eruption rate by the cylindrical ash injection volume, the particle concentration $C$ at each mesh point can be calculated for each time step during the release of ash.

   Ash is settling out of the eruption cloud dependent on its grain size. Summarizing the results of Paladio-Melosantos et al.

(1996) and Wiesner et al. (1995) in an area of about 600 km around the volcano, ash fallout ranges from -4 to 9 $\phi$ (16 to 0.00195 mm). Since large particles below 1 $\phi$ settle too fast and would require very small time steps and very small particles above 8 $\phi$ sediment outside the simulation domain, we neglected these particle sizes. For a sake of simplicity, we used a Gaussian distribution around a mean grain size of 4.5 $\phi$ with a standard deviation of $\sigma_\phi = 2.5$ which corresponds well with the estimated bulk mean grain size of 4 $\phi$ of the Pinatubo tephra deposit (Fero et al., 2009). The grain size categories used for the

estimation of the settling velocities are listed together with the particle diameter and the particle density in Table 3.

### 3.2.2   Mesh generation and refinement settings

The initial mesh consists of three cubes, each extending from 4.5°N to 24.5°N and from 0 to 23km height and from 69.5°E to 89.5°E (cube 1), 89.5°E to 109.5°E (cube 2) and 109.5°E to 129.5°E (cube 3). Each cube contains 6 tetrahedrons, so initially there are 18 tetrahedrons in the domain. Due to the minimum refinement level (*Coarse Mesh Level* in Tab. 2) each of these

tetrahedrons is refined 7 times resulting in a total of 2304 tetrahedrons in the complete modeling domain. The *Coarse Mesh Level* gives the level of global and uniform refinement at the initialization of the grid and the maximum level to which an




| Grain size [$\phi$] | Diameter [mm] | Density [kg/m$^3$] | Wt.% |
|---|---|---|---|
| 2 – 1 | 0.25 – 0.5 | 1430 | 4.17 |
| 3 – 2 | 0.125 – 0.25 | 1720 | 11.34 |
| 4 – 3 | 0.0625 – 0.125 | 2010 | 20.66 |
| 5 – 4 | 0.0313 – 0.0625 | 2300 | 25.23 |
| 6 – 5 | 0.0156 – 0.0313 | 2300 | 20.66 |
| 7 – 6 | 0.0078 – 0.0156 | 2300 | 11.34 |
| 8 – 7 | 0.0039 – 0.0078 | 2300 | 4.17 |

**Table 3.** Particle diameters and densities utilized in the sedimentation simulations. The particle densities as a function of the grain size listed here are extracted from Macedonio et al. (1988).

| Mesh refinement | min. horizontal edge length | min. vertical edge length |
|---|---|---|
| Refinement level 8 | 4.5 degree | 5.75 km |
| Refinement level 17 | 0.6 degree | 0.7 km |

**Table 4.** Minimum edge length of the tetrahedrons in horizontal and vertical direction.

element is coarsened, whereas the *Fine Mesh Level* defines the maximum level to which an element can be refined. After the initial refinement of the mesh (corresponding to a mesh level of 8), the minimum length of the edges of the tetrahedrons in horizontal direction is 4.5 degree. With the maximum refinement level of 17, minimum edge length of 0.6 degree in horizontal and 0.7 km in vertical direction are achieved (compare Tab. 4).

In Figure 2, the structure of the three dimensional adaptive tetrahedral mesh of the ash cloud on June 15 23:30 PDT is displayed. The cloud is seen form the side. The mesh is composed of tetrahedrons with variable size. Please note that particle settling is not considered in this simulation. On the right side — where the mesh resolution is higher — the ash emissions are inserted into the model (see above), leading to a larger gradient in the ash concentration which in turn starts the mesh refinement dependent on the refinement and coarsening tolerances (see Tab. 2 and section 2). An animation of the advection of

the three dimensional cloud can be found in the supplementary material.

    The impact of the maximum level of refinement (*Fine Mesh Level*) is demonstrated in Figure 3 where a horizontal cross section at the height of the mean transport level (17 km) is shown. Between the ash clouds simulated with a *Fine Mesh Level* of 14 (Fig. 3a) and a *Fine Mesh Level* of 17 (3b), significant differences in shape and ash concentration of the clouds can be observed. When comparing the ash concentration for different mesh resolutions, it is import to consider that the total amount

of ash inserted initially is slightly different for the different refinement levels, because the discrete volume of the cylinder in which the ash is inserted is dependent on the cell size. Using a *Fine Mesh Level* of 14 the discrete volume amounts to 97.6 % of the analytical volume of the cylinder (compare section 3.2.1) while with a *Fine Mesh Level* of 17 the inserted ash mass already accounts for 99.5 % of the original ash mass.





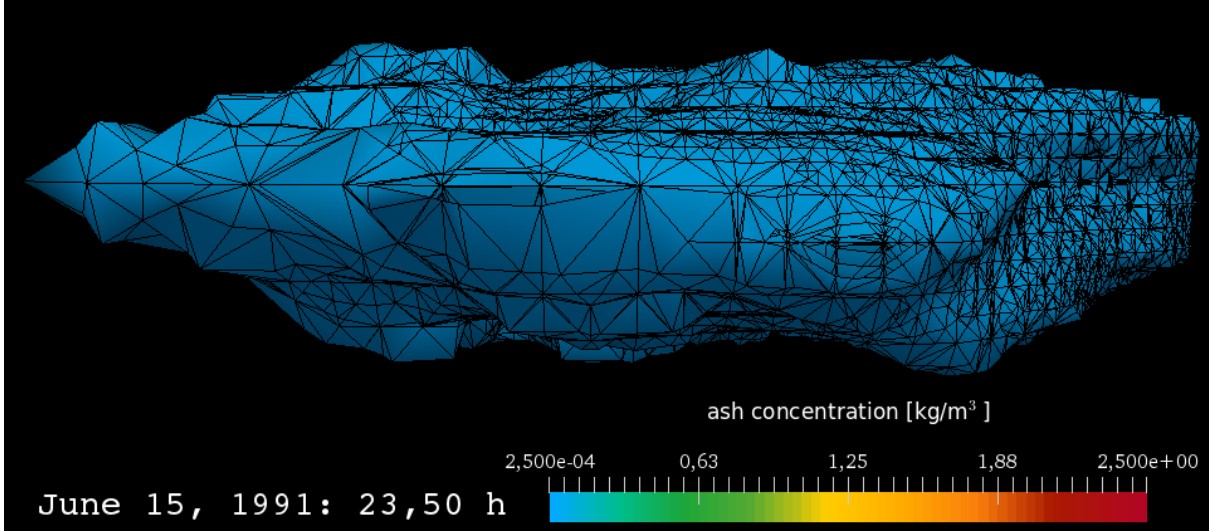

**Figure 2.** Triangulated mesh on the surface of the ash cloud. The ash cloud is represented on June 15 at 23:30 in a side view (from south). The concentration on the surface of the ash cloud is displayed in kg/m³.

For *Fine Mesh Level* larger than 16, we found that the results do not change significantly any more apart from small differences in the ash concentration, but the general behavior of the ash cloud is preserved. Hence, from here on we utilize a *Fine Mesh Level* of 17 as maximum refinement level allowing for fast computation of the ash spreading.

### 3.3   Results for the standard model setup

Figure 4 shows the evolution of the settling ash cloud for a particle size of 4.5 $\phi$ (0.0469 mm) (animation in the supplement). At 10:00 on June 13, the ash cloud is centered above the volcano at a mean height of around 16 km (Fig. 4a). In the following four hours, the ash cloud is settling down to a medium height of around 13 km and is slightly advected to the west (Fig. 4b). On June 14 14:00, ash particles of the eruption on June 13 have sedimented down to ground, while the eruption cloud from the second eruption phase starting on June 14 13:09 (compare Tab. 1) is still close to its initial position (Fig. 4c). One day

later on June 15 at 14:00, shortly after the onset of the climactic eruption, the *ash column* from the second eruption phase is centered above the South China Sea. While settling, ash particles are advected to the southwest, especially between heights of 8 to 10 km (Fig. 4d). In the following hours, the ash cloud is drifting further in southwest direction (Fig. 4e and 4f). After the last eruption phase ended (on June 16 10:41), the ash cloud is sinking down and is advected to the west-southwest (Fig. 4g and 4h).

Figure 4 only shows ash concentration on the surface of the ash cloud. Figure 5 allows a look into the ash cloud where cross sections of the iso-surfaces of the ash concentration on June 14 at 14:00 and June 15 at 22:00 are displayed. When ash is inserted, the ash concentration inside the cloud is initially homogeneous (Fig. 5a). While the ash is advected and sediments, ash particles are dispersed and the ash concentration decreases. Since ash is inserted continuously between a height of 14 to





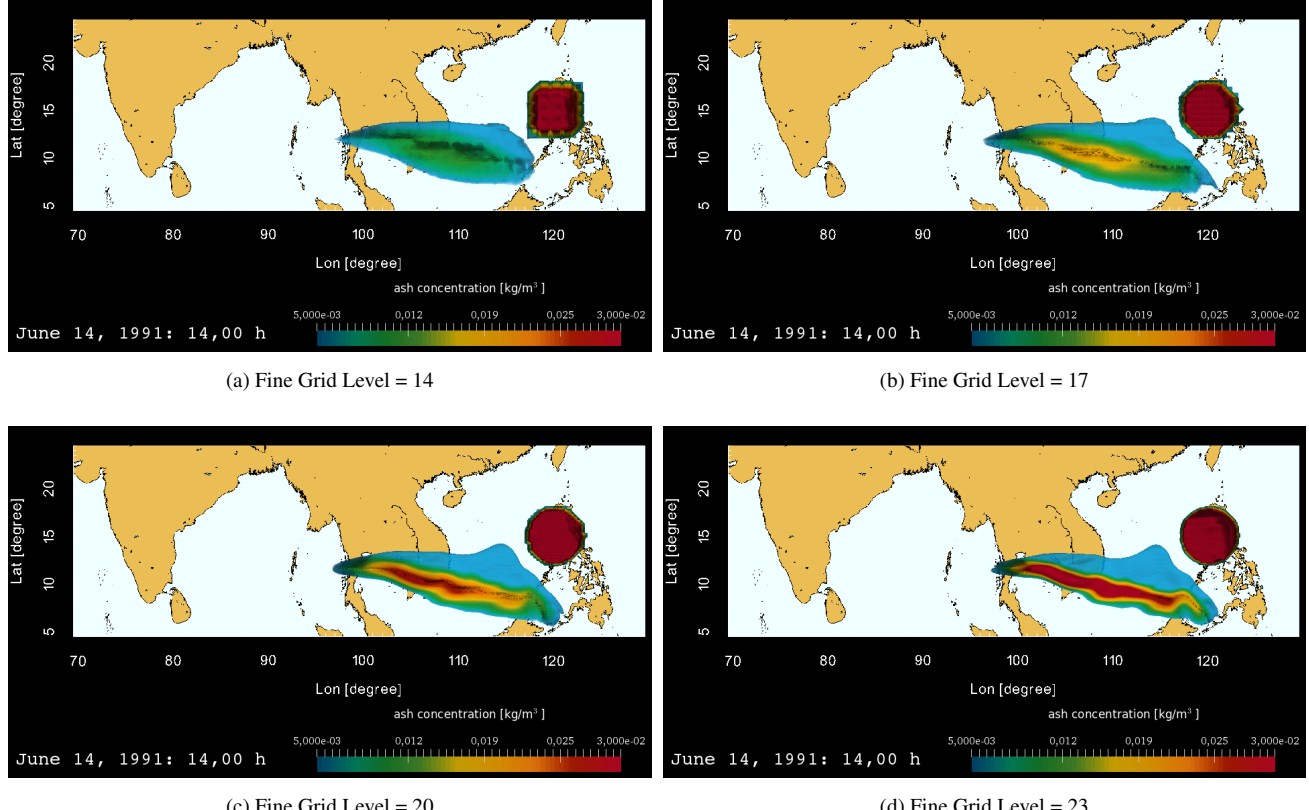

**Figure 3.** Horizontal cross section of the ash cloud at a height of 17 km, modeled with a *Fine Mesh Level* of 14, 17, 20 and 23. Results for June 14 at 14:00 PDT are shown. The colors indicate ash concentration in kg/m$^3$.

km, the highest ash concentrations are obtained in the upper part of the cloud (Fig. 5b). After the onset of the climactic eruption, the ash concentration inside the cloud significantly increases (note that different colorbars are used).

The results of the advected and sedimented ash cloud on June 15 at 22:00 are displayed for model runs with different grain sizes and for a simulation without the settling of particles in Figure 6. In the simulation without the settling of particles (Fig.
5  6a), most of the ash is advected in the stratosphere and upper troposphere in west- and southwestward direction. The larger the ash particles are, the higher is the settling velocity and the more is the ash advected to the south due to the changing wind patterns in the atmosphere. For particles with grain sizes of 7.5 $\phi$ and 6.5 $\phi$, advection dominates and the ash cloud is drifting more or less horizontally to the west-southwest (Fig. 6b and 6c). The effect of sedimentation becomes visible for particle sizes larger or equal to 5.5 $\phi$ (Fig. 6d). For simulations with particle sizes between 4.5 and 3.5 $\phi$, a certain amount of the ash particles
10  sedimented to ground on June 15 22:00, but advection still has an impact on the motion of the cloud (Fig. 6e and 6f). For even larger particles, sedimentation dominates the advection of ash particles and the particles are settling down in a nearly vertical ash column (Fig. 6g and 6h).





(a) June 13 10:00

(b) June 13 14:00

(c) June 14 14:00

(d) June 15 14:00

(e) June 15 18:00

(f) June 15 22:00

(g) June 16 14:00

(h) June 17 14:00

**Figure 4.** Settling of the ash cloud over time for particles with a grain size of $4.5\,\phi$. The concentration on the surface of the ash cloud is displayed in kg/m$^3$. Note that different colorbars are used.





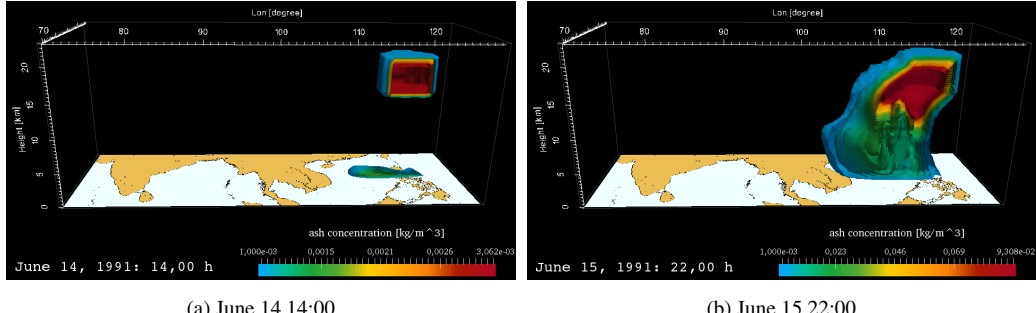

(a) June 14 14:00                    (b) June 15 22:00

**Figure 5.** Ash concentration of the ash cloud on June 14 at 14:00 and on June 15 22:00 for particles with a grain size of 4.5 $\phi$. A cross section of isosurfaces of the ash concentration is displayed in kg/m$^3$. Note that different colorbars are used.

Projecting the extend of the calculated ash cloud onto satellite observations made during the Pinatubo eruption (see section 3.1) is also quite instructive to determine which particle size or sizes are the best to achieve a good fit with observations. In Figure 7a, the result of a simulation without the settling of particles is shown. The winds in the lower stratosphere advect most of the ash to the west and only a very small amount of ash travels to the southwest. For particles with a grain size of 5 $\phi$ (0.0313 mm), the ash settles slowly to lower altitudes, where the wind is directed southwestwardly. The particles are more or less evenly distributed between west and southwest (Fig. 7b). With increasing particle size, ash is advected in a heart-shaped cloud to the west and the southwest (Fig. 7c and 7d). The simulated ash cloud with particles of a grain size of 4 $\phi$ (0.0625 mm) is of a much smaller extent, because the particles are settling significantly faster and the advection of the cloud by wind plays a minor role (Fig. 7d).

Comparing all simulations with different settling velocities, the best fit to the data of Lynch and Stephens (1996) is obtained by simulations with a particle size of 5 $\phi$. The agreement between the simulated ash cloud and the outline of the umbrella cloud identified by Lynch and Stephens (1996) increases, when the results for particle sizes of 4, 4.5 and 5 $\phi$ are combined (Fig. 8). An animation of the compared results can be found in the supplementary material. On June 15 at 22:30, the simulated ash cloud matches the data of Lynch and Stephens (1996) very well (Fig. 8a). However, at 4:30 on June 16, the modeled ash cloud and the observed contour of the ash cloud generally agree with each other, but the southern extension of the umbrella cloud could not be reproduced (Fig. 8b).

## 4   Discussion

In this study we adapted an existing adaptive semi-Lagrangian advection model to model the dispersion of volcanic ash emissions including the sedimentation of particles from the ash cloud. We matched our results with published satellite data of the umbrella cloud during the climactic phase of the Pinatubo eruption and found the simulation of the advection and the sedimentation to match observations quite well. The best fit between modeled and observed data were obtained by combining results







(a) no settling

(b) $\phi = 7.5$

(c) $\phi = 6.5$

(d) $\phi = 5.5$

(e) $\phi = 4.5$

(f) $\phi = 3.5$

(g) $\phi = 2.5$

(h) $\phi = 1.5$

**Figure 6.** Modelled ash cloud for the grain size categories listed in Table 3 and for a simulation neglecting the settling of particles. The concentration on the surface of the ash cloud is displayed in kg/m$^3$. Note that different colorbars are used.





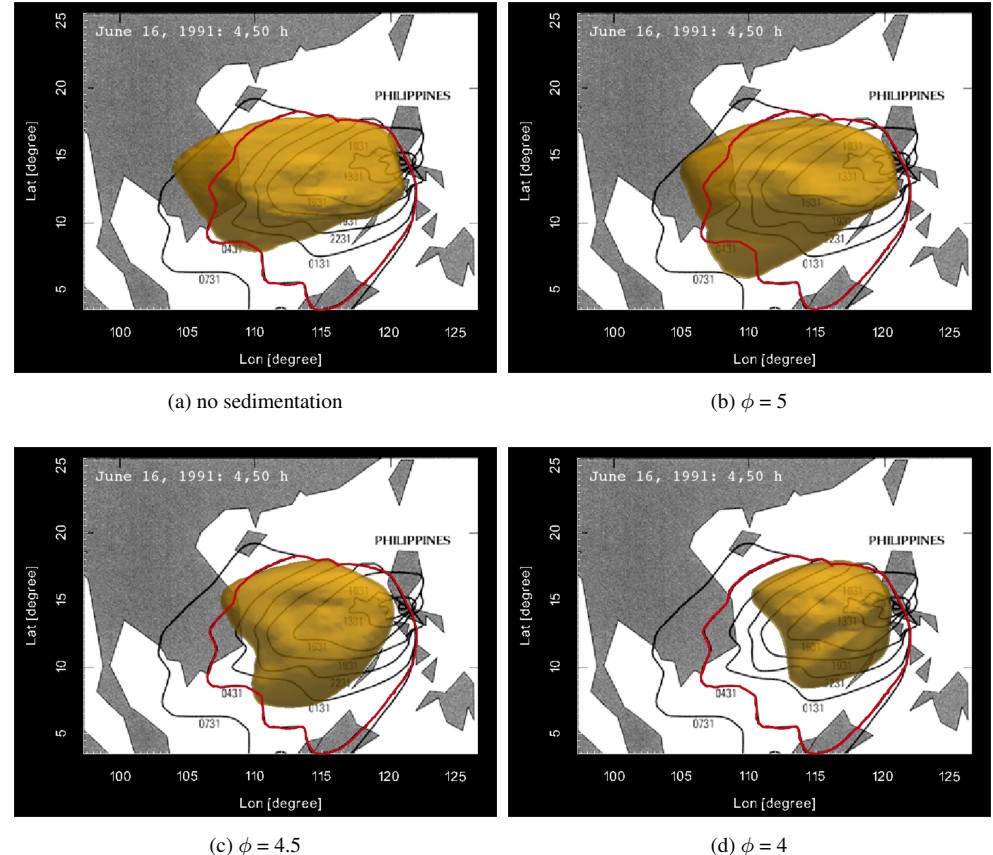

**Figure 7.** Advected ash cloud on June 16 4:30 for particle sizes of 4, 4.5 and 5 $\phi$ and for a simulation without the settling of particles. In the background, the outer edge of the observed ash cloud is displayed in three hour intervals (Observed contours from: Lynch and Stephens (1996)). The outer edge of the simulated ash cloud was defined as the isosurface with an ash concentration of 0.05 kg/m$^3$, producing the closest fit to the observed data.

from simulations with multiple particle sizes (Fig. 8b). In the following we will first test the sensitivity of our results to some of the main input parameters before we turn to a discussion of the performance advantage of our model compared to fixed grid models.

## 4.1 Sensitivity study

5  Table 2 lists various input parameters used in the model calculations. Since the initial radial expansion of the Pinatubo cloud is not included in our model, the initial radius of the ash cloud was chosen to be relatively large. In the models of Fero et al. (2009), an initial radius of around 400 km was found to be necessary to account for the radial expansion. We varied the radius between two and five degrees (approximately 222 to 555 km). Following Fero et al. (2009), most of the ash was advected at a height of around 17 km, but some amount of the ash was injected at much higher altitudes (Holasek et al., 1996). We therefore





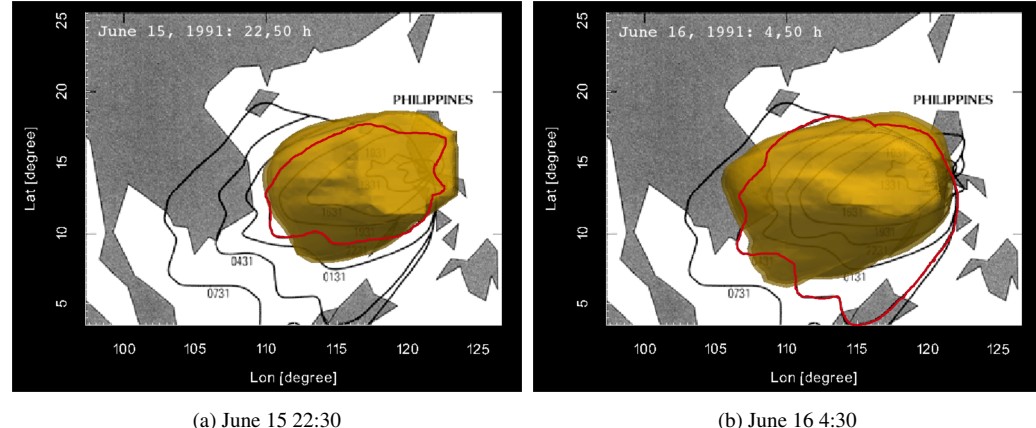

(a) June 15 22:30                                      (b) June 16 4:30

**Figure 8.** Combined results for particle sizes of 4, 4.5 and 5 $\phi$ and the outline of the umbrella cloud identified by Lynch and Stephens (1996).

tested medium cloud heights between 15 and 21 km. Koyaguchi (1996) reported, that the thickness of the umbrella cloud was between 3 to 5 km, while Self et al. (1996) mentioned a cloud thickness of 10 to 15 km. We therefore varied the umbrella cloud thickness between 2 to 8 km. Varying the cloud thickness between 2 and 8 km height did not impact the ash dispersion significantly so this is not discussed in further detail below. In the following sensitivity analysis we use a medium grain size of

4.5 $\phi$ (0.0469 mm) in order to not obscure the results by the differences in settling velocities.

The influence of the initial cloud radius is shown in Figure 9. The ash cloud was simulated with an initial cloud thickness of 6 km, a medium height of 17 km, a grain size of 4.5 $\phi$ and an initial radius of 2, 3, 4 and 5°. The area covered by the cloud decreases with a decreasing radius and the shape of the modeled ash cloud becomes more circular for larger radii (compare Fig. 9a and 9d). At 22:30 on June 15, the best fit to the outline of the observed ash cloud (identified by Lynch and Stephens

(1996)) is obtained with an initial cloud radius of 4°.

Figure 10 shows the effect of variation in the initial mean cloud height on the extent of the simulated ash cloud. The higher the ash is inserted, the more the cloud is advected to the west by stratospheric winds (Fig. 10d). At lower altitudes, wind mainly carries the ash in southwest direction and the ash cloud covers a heart-shaped area (Fig. 10a). The modeled ash clouds with an initial height of 17 and 19 km (Fig. 10b and 10c) better matches the contour of the ash cloud identified by Lynch and Stephens

(1996), but the expansion of the umbrella cloud to the south is not reproduced.

Since non of our model simulations, even the best fit one shown in Figure 9, do match the observations completely — in particular in the south — we attribute the remaining differences between the modeled and the observed ash dispersion to the following facts:

1. The initial conditions of the eruption cloud are not well-known, including the vertical mass distribution in the plume.

2. The radial expansion of the umbrella cloud is accounted for by a larger initial cloud radius, which might induce deviations in cloud shape.





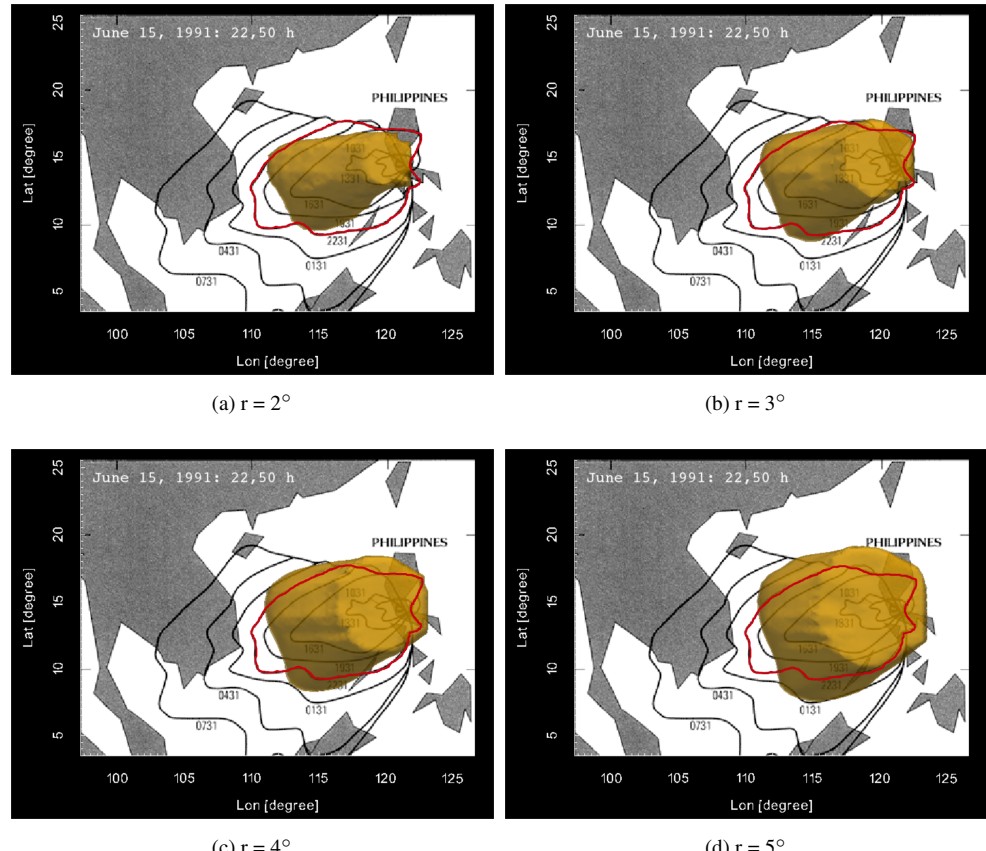

(a) r = 2°          (b) r = 3°

(c) r = 4°          (d) r = 5°

**Figure 9.** The modelled ash cloud on June 15 22:30 with an initial radius of 2, 3, 4 and 5°and the outer edge of the observed umbrella cloud in three hour intervals (observed outlines from: Lynch and Stephens (1996)). The modelled ash cloud is displayed for June 15 22:30.

3. The pre-calculated wind field might not reproduce correctly the conditions during the eruption, especially the pass by of Typhoon Yunya.

4. The heavy rain caused by Typhoon Yunya is neglected in this model simulations, so that wash-out and vertical transport might be underestimated.

5. Uncertainties in the outline of the umbrella cloud identified by Lynch and Stephens (1996). Satellite observations only reflect the outer contours of the ash cloud in the upper most layers. In addition, ash clouds are hard to distinguish from meteorological clouds.

## 4.2 Performance due to adaptive meshing

The main advantage of this model is the use of an adaptive tetrahedral mesh leading to significantly reduced computational costs while tracking the volcanic emissions with a very high local mesh resolution. In order to determine the advantage of our





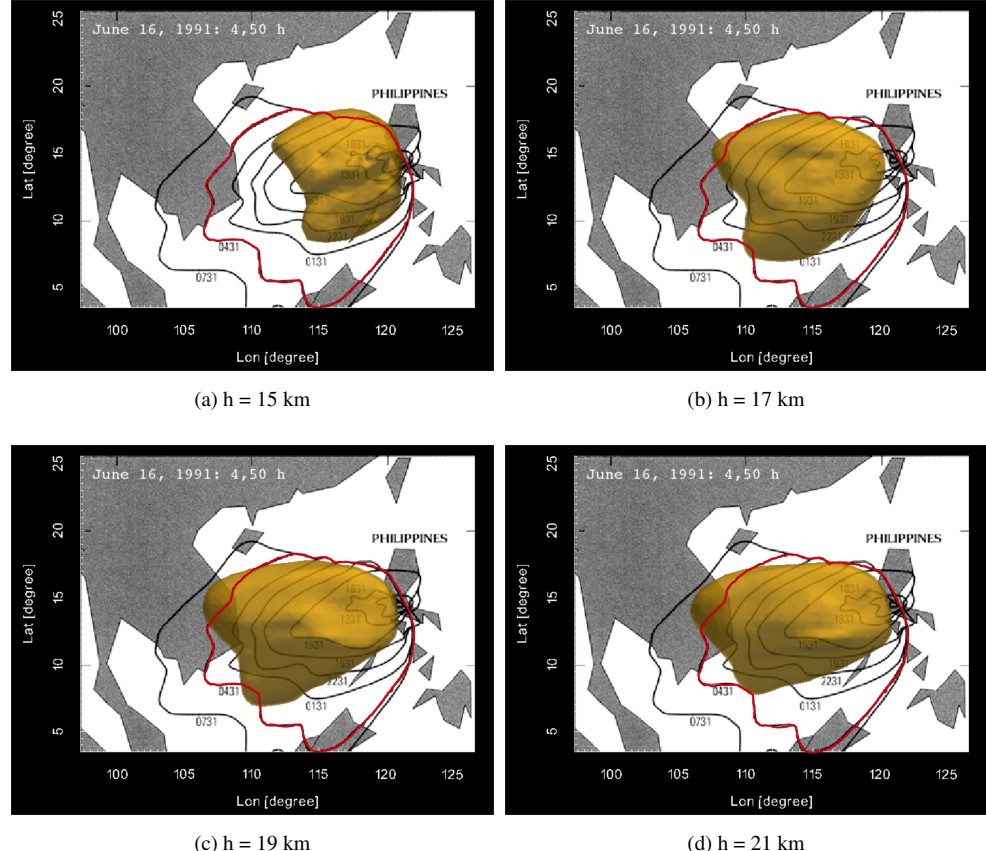

    (a) h = 15 km                                   (b) h = 17 km

    (c) h = 19 km                                   (d) h = 21 km

**Figure 10.** Advected ash cloud on June 16 4:30 inserted at a mean height of 15, 17, 19 and 21 km in combination with the extent of the observed ash cloud analyzed by Lynch and Stephens (1996).

adaptive meshing approach compared to fixed grid calculations we carried out a series of model runs with a fixed grid (i.e. we set the fine grid level equal to the coarse grid level to achieve a fixed grid) and compare them to our standard run with an adaptive mesh. In this case study, all simulation are carried out without the sedimentation of particles. In Figure 11, results of simulations on an uniform grid with refinement levels of 13, 14 and 15 are compared to a result of a calculation on the adaptive

5   mesh. Obviously, the shape of the ash cloud is recovered quite well in all calculations, but small patterns in the shape and the ash concentrations are much better recovered with the finer grid structure. As mentioned above (section 3.2.2), the initial ash mass varies slightly for different mesh resolutions.

    In Fig. 12 we compare the computational costs of the different model calculations. The simulation on the adaptive mesh needed only about 50 min, while the calculation on an uniform mesh with a fine grid level of 17 (i.e. the same maximum local

10  resolution as in our adaptive mesh calculation) required already around 9 h. Those significantly reduced computational cost would allow for ensemble runs with varying meteorological boundary conditions as well as different ash injection assumptions to better constrain and forecast probable dispersion patterns and directions (see e.g. (**?**)).





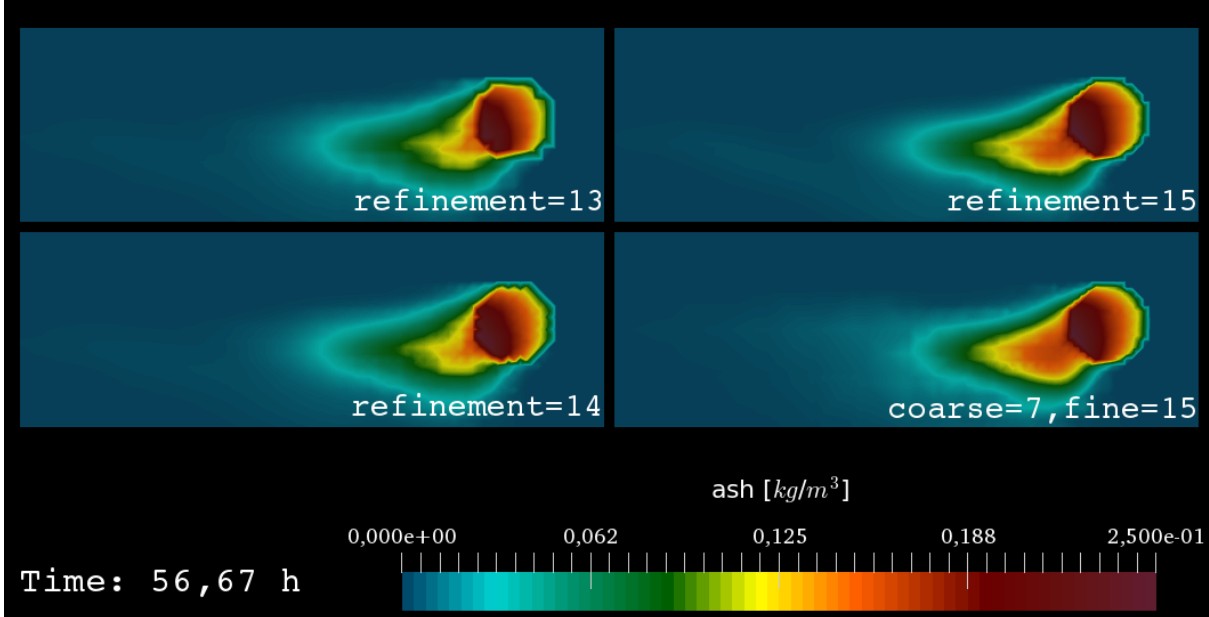

**Figure 11.** Comparison between simulations on an uniform mesh with a refinement level of 13 (top left), 14 (lower left) and 15 (top right) and a simulation on the adaptive mesh with a coarse mesh level of 8 and a fine mesh level of 17 (downright). Cross sections at a height of 18 km on June 17 at 13:20. The same figure including the mesh structure can be found in the online supplement.

## 5    Conclusions

In this study we have demonstrated the versatility of adaptive meshing algorithms for modeling the dispersion of volcanic emissions. Especially the high performance of this code would allow, if implemented into operational ash dispersion models, a significant improvement of dispersion predictions as model runs could be carried out significantly faster compared to codes using a fixed grid. The research community benefits from such a faster code by being able to resolve the fine filamented structure of volcanic emissions during their transport as well as test more boundary conditions, newly developed sedimentation models (e.g. (Bagheri and Bonadonna, 2016)) and complex chemical reactions which could occur between different trace gases in the atmosphere while being transported (e.g. (Hoshyaripour et al., 2015))

In our sensitivity study we have shown that the initial conditions of the ash cloud significantly influence the region impacted by the ash cloud. Even in cases where meteorological predictions, the initial height, the extent of the ash cloud as well as the mean grain size of the erupted particles are not very well constrained, our model could be used for forecasting the advection and the sedimentation of ash after a volcanic eruption through ensemble runs and thereby contribute to assessment and mitigation of risks, posed by drifting ash clouds.

A further application of the model is to predict the ash loading on the Earth's surface from tephra fallout which only needs an additional two-dimensional array to sum up the deposited ash.




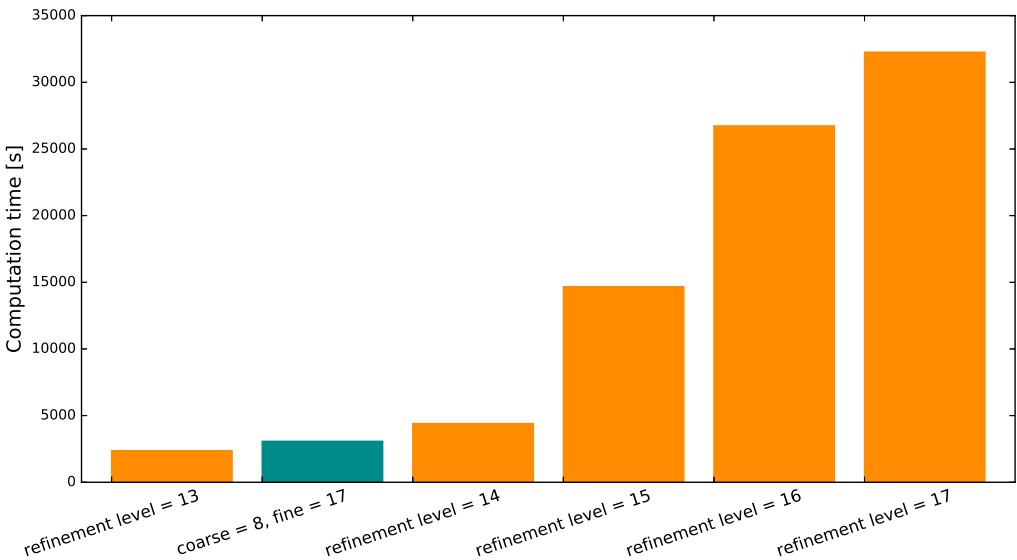

**Figure 12.** Comparison of the computation times of simulations on an uniform mesh with refinement levels of 13, 14, 15, 16 and 17 and on the adaptive mesh with a coarse mesh level of 8 and a fine mesh level of 17. Calculations were carried out on a lenovo thinkpad with an i3-2310M processor and 8 GB of main memory.

**Appendix A**





| Date 1991 | Time [PDT] | Column Height [km] | Date 1991 | Time [PDT] | Column Height [km] |
|---|---|---|---|---|---|
| June 13 | 08:41 | 24.0 | June 15 | 13:41 | 37.5 |
| June 14 | 13:09 | 21.0 | | 14:41 | 40.0 |
| | 13:41 | 22.5 | | 15:41 | 38.0 |
| | 14:10 | 15.0 | | 16:41 | 32.0 |
| | 15:41 | 19.0 | | 17:41 | 34.5 |
| | 18:53 | $\geq$ 24.0 | | 18:34 | 35.0 |
| | 19:41 | 20.0 | | 19:41 | 29.0 |
| | 22:18 | 5.0 | | 20:41 | 28.0 |
| | 23:20 | 21.0 | | 21:41 | 27.0 |
| | 23:30 | $\geq$ 21.0 | | 22:31 | 26 – 28 |
| | 23:41 | 18.0 | | 22:41 | 26.5 |
| June 15 | 01:14 | 23 – 25 | | 23:41 | 22.5 |
| | 01:41 | 21.5 | June 16 | 01:41 | 20.0 |
| | 03:41 | 20.5 | | 02:41 | 19.0 |
| | 05:55 | 12 – 20 | | 03:41 | 17.5 |
| | 06:34 | 20.5 | | 04:41 | 16.0 |
| | 08:10 | 12 – 20 | | 05:41 | 14.0 |
| | 08:41 | 17.5 | | 06:41 | 14.0 |
| | 10:27 | > 20 | | 07:41 | 14.0 |
| | 10:41 | 21.5 | | 08:41 | 13.0 |
| | 12:13 | 8.0 | | 10:41 | 15.0 |
| | 12.34 | 24.5 | June 16 – 18 | | 0.2 – 19 |

**Table A1.** Chronology of eruption column heights between June 13 06:00 and June 18 00:00. Data compiled from Holasek et al. (1996) and Self et al. (1996)

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
