# Peer review of "An Adaptive Semi-Lagrangian Advection Model for Transport of Volcanic Emissions in the Atmosphere"

_Natural Hazards and Earth System Sciences, 2017_

## Referee Comment (RC1) · A. Folch (Referee) · 3 Jun 2017

Comments on "An Adaptive Semi-Lagrangian Advection Model for Transport of Volcanic Emissions in the Atmosphere" by Gerwing et al.

Summary

This manuscript presents a semi-Lagrangian atmospheric dispersal model adapted to volcanic ash. The model is solved using a Finite Element Method (FEM) using grid adaptivity on a tetrahedral unstructured mesh. The 1991 Pinatubo climactic phase is used as an application example, with some qualitative model validation. The mod-

elling strategy (novel in the context of tephra dispersal models) may have potential advantages. However, my overall impression is that several aspects of this manuscript should be clarified/better explained before accepted for publication.

General comments

1. The model does not take into account for processes known to be relevant in ash cloud dynamics such as diffusion, dry/wet deposition mechanisms or ash aggregation. As a result, the model cloud dynamics limits to wind advection and particle sedimentation. Near-source effects (e.g. plume dynamics or gravity current) are ignored. On the other hand and more important, I do not understand why particle ground deposition is not contemplated. A part from the obvious interest of estimating ash/tephra fallout, the computation of the deposit is necessary (together with satellite imagery) for model validation (see below)

2. The innovative aspect of this manuscript is the use of adaptive grids in ash cloud simulations. However, several aspects regarding the numerical algorithm and its advantages are not detailed. Is the mesher embedded in the code? How often is the refinement applied? At each time integration step or at user-defined time intervals? The overhead of performing variable interpolation after each refinement and weather this is done in a conservative way or not is not mentioned.

3. The authors conclude that (pg 18, lines 1-4): "we have demonstrated the versatility of adaptive meshing algorithms for modeling the dispersion of volcanic emissions. Especially the high performance of this code would allow, if implemented into operational ash dispersion models, a significant improvement of dispersion predictions as model runs could be carried out significantly faster compared to codes using a fixed grid". Certainly there is a potential, but I think that this conclusion is too precipitated for many reasons. Constraining to Eulerian models (where this assertion would make sense), an important point not mentioned is code parallelism (most operational models actually run in parallel). The drawbacks of mesh adaptivity in the parallel execution

of transient (time-evolving) problems are well-known: redefine the optimal domain de-
composition at each refinement to ensure processor load balance has large associated
interpolation and communication costs, code scalability breaks, etc. As a result, it is
unclear whether the strategy would suppose a gain or not when running on hundreds
of processors. This affects the main conclusion of this manuscript.

4. Another controversial aspect which is not discussed by the authors is that "we apply
the model for individual particle diameters and then combine the results of different runs
to predict the sedimentation of the complete grain size distribution", pg. 4, line 22). I
understand that this is necessary because the different particle sizes require of different
refinements. However, how this affects efficiency in case on several (e.g. 10) particle
classes is not even mentioned. Is the comparison shown in Figure 12 for a single
class? This may not be fair, but it is difficult for a reader to extract conclusions since
no details are given on the fraction of computation time of remeshing/interpolation. I
suggest comparing the time of 10 simulations (coarse 8, fine 17) with that of a single
run with all 10 classes and fixed grid.

5. I missed details on the model numerical algorithm. Is it explicit or implicit? What
about the time integration step (only 10 min is mentioned in pg. 4 line 17, but based on
what?).

6. Model validation could certainly be improved. I wonder why the Pinatubo erup-
tion was selected to this purpose given some obvious difficulties: the role of the
gravity current (see Costa et al., Geophysical Research Letters, vol. 40, 1–5,
doi:10.1002/grl.50942, 2013), the particular meteorological conditions at that time, the
lack of extensive deposit sampling and inferred TGSD, etc. In any case, some quan-
titative model validation would be worth. On the other hand, it is stated (e.g. Table 4)
that the refinement level goes down to <5 km. However, it seems that the driving me-
teorology is at 55 km and the wind field is linearly interpolated. Does it make sense?
My impression is that refinement at sharp concentration gradients helps because it
reduces numerical diffusion. . .

7. Sensitivity study (section 4.1). The effect of variation in initial cloud height is actually a combined effect of injection height and driving meteorology (REMOTE)...

Specific comments

Pg. 2, line 16. Parenthesis in reference

Pg 2, line 26. "very low computational cost". This is too vague and generic...Also, values of "seconds" is not what Fig. 12 shows.

Pg. 3, lines 8-9. The resolution of 0.5x0.5 is that of REMOTE? If so, I do not understand which is the gain with respect to driving global ECMWF (Era-Interim?) data, already available at this resolution. Do you mean that the mesoscale simulation is not used to increase the wind field resolution?

Pg. 3, line 12. Parenthesis in reference

Pg. 3, line 12. Like?

Pg. 3., line 15. Model → domain

Pg. 3, equations (1) and (3). Even if only advection is considered, shouldn't these equations include the terms $C(div\_u)$ (where $u = u\_REMOTE+u\_t$)? Is the wind from REMOTE divergence free? Can the z-gradient of the terminal settling velocity be ignored?

Pg. 4, line 26: "Since this work is a first case study of the modeling of sedimentation of ash particles on an adaptive mesh the impact of rain on the sedimentation and aggregation of ash particles is neglected." This is ok, but I have concerns about how aggregation could ever be incorporated in a future using this strategy. With sedimentation each particle class has to have its own mesh (different model runs) and aggregation requires concurrency.

Pg. 5, lines 12-13. Are hours correct?

Pg. 6, line 5: "This atypical wind in the lower and middle troposphere caused the wide distribution of tephra in nearly all directions around the volcano". Was this a meteo effect or because the radial gravity current?

Pg. 8, Table 4. The horizontal/vertical element aspect ratio seems very large for tetrahedral elements ($\sim$100). Any hint on mesh element quality? In case of small angle, can this lead to oscillations/convergence problems?

Pg. 9, line 15. "concentration on the surface". What does it mean? Which value? How is this defined?

Figure 7 (and others). Why so many contours of observations? Only the corresponding to the time should be shown for clarity.

---

## Referee Comment (RC2) · M. Herzog (Referee) · 14 Jul 2017

The paper presents simulations of the dispersion of the Pinatubo ash cloud using a transport model with adaptive mesh und prescribed winds. The application of adaptive grid methods to tracer transport problems is not new, however, the application of the chosen semi-Lagrangian method to a volcanic plume and the inclusion of particle sedimentation has – to my knowledge - not been done before. As such the paper warrants publication. However, there are number of issues that need to be addressed before publication.

The biggest issue is the model initialization and forcing. It is said (page 7, line 6-8)

that the initial concentration is derived by dividing the eruption rate by the injection volume. However, this results in a flux of ash mass per unit volume and time, not a concentration. This flux should have been maintained over the duration of the corresponding eruption phase. However, it seems that an initial perturbation/concentration was used instead. If this is true, then this is a major flaw of the paper. Only adding the ash emitted within one second underestimates the erupted ash mass by many orders of magnitude. Releasing all ash erupted during each phase (as listed in table 1) instantaneously at the beginning of an eruption phase is equally unrealistic. This issue needs to be addressed, simulations repeated if needed before the manuscript can be published.

Also, the total amount of ash should not depend on the choice of the grid. If the forced volume is different, then the ash flux into that volume should be (slightly) different to compensate.

In the abstract (page 1, line 3) and conclusions (page 18, line 5) the authors state that adaptive meshes are useful to resolve filament structures of volcanic emissions. However, in the chosen example and in the presented results no filaments are present. The authors need to better justify and motivate the selection of the Pinatubo eruption as a case study.

The model description (chapter 2) does not describe the semi-Lagrangian transport model in any detail. The reference (Behrens, 1996) is given but it is not clear from the text that the transport model is described in there. More details at least about the model concept are needed with a proper reference to the Behrens (1996) paper for further details.

Add in the last paragraph of page 4 that ash is treated as a passive tracer.

Particle deposition (page 4, line 28) was not monitored. It is unclear why, when it would have been as easy to implement as suggested in the conclusions (page 18, line 14-15). Since deposited ash is the main source of information for historical eruptions it would

have been good to test the resolution dependence of that deposition.

The mass eruption rate in table 1 has wrong units. I would assume it is kg per second instead. In table 2 add 'refinement' to the word level to avoid confusion with vertical levels. A cloud radius in degrees is an odd choice since this means an elliptical shape in physical space. This is inconsistent with the stated initial radial expansion of the plume on page 5. Discuss and clarify.

I disagree that results are converged for fine mesh levels larger than 16 (page 9, line 1-2). According to figure 3, ash concentrations in the centre of the plume to the south west of the volcano increase significantly from fine mesh levels 17 to 20 and to 23. Quantify the differences and discuss convergence or non-convergence in greater detail.

Other minor issues include:

Page 1, line 17: fall **out** of tephra

Page 1, line 18: tephra fall(s) **out** can lead...

Page 2, line 2-3: add: ...warmer winters and colder summer on the Northern hemisphere continents through dynamical feedbacks and radiative forcing, respectively (Robock, 2000).

Page 2, line 4-6: timescales of minutes don't influence the diurnal cycle

Page 5, line 3: remove 'one of' since it has been said before that Pinatubo was the largest eruption in the 20th century.

Page 6, line 5: what is 'atypical' about the winds?

Page 7, line 20: say already here that 7 times means refinement level 8 (in table 2). Write 'in the initial model domain before refinement'.

Page 10, figure 3: increase font of colour bar and text.

Page 11, figure 4: use identical and more meaningful colour bar. There are no yellow or red colours visible. What defines the surface of the ash cloud? If it is a threshold concentration the figure should show an iso-surface. Explain.

Page 12, figure 5: use same colour bar for both panels to enable comparison.

Page 12, line 15: delete 'however'.

Page 12, line 11-17: use information from table 3 for superposition of different ash sizes. Ideally, this should give the best fit and allow for a more independent validation.

Page 13, figure 6: use identical and more meaningful colour bars. Yellow and red colours not visible.

Page 15, line 16: 'since non**e** of our (model) simulations'

Page 16, line 3-4: write: will (not might) be underestimated

Page 17, line 5-7 and page 18, figure 11: it is not obvious to me that the shape is recovered well in all calculations. Quantify differences, in particular, discuss differences between top right and bottom right panels (identical fine resolution). Label for bottom right panel: shouldn't it read 'coarse=8'?

Page 17, line 8-12: this is the common way to calculate performance gains due to adaptation. However, a transport model written and optimized for constant resolution can be significantly faster than an adaptive grid code run at constant grid resolution. Discuss to which extend this issue might apply here.

Page 17, line 12: unresolved reference

---

## Author Comment (AC1) · 29 Aug 2017

1. The biggest issue is the model initialization and forcing. It is said (page 7, line 6-8) that the initial concentration is derived by dividing the eruption rate by the injection volume. However, this results in a flux of ash mass per unit volume and time, not a concentration. This flux should have been maintained over the duration of the corresponding eruption phase. However, it seems that an initial perturbation/concentration was used instead. If this is true, then this is a major flaw of the paper. Only adding the ash emitted within one second underestimates the erupted ash mass by many orders of magnitude. Releasing all ash erupted during each phase (as listed in table 1)

instantaneously at the beginning of an eruption phase is equally unrealistic. This issue needs to be addressed, simulations repeated if needed before the manuscript can be published. Also, the total amount of ash should not depend on the choice of the grid. If the forced volume is different, then the ash flux into that volume should be (slightly) different to compensate.

The emitted ash per eruption rate was neither released instantaneously at the beginning of the eruption nor was only the ash amount emitted during one second introduced per time step. The calculated mass eruption rates in kg/s (listed in Table 1 in the manuscript) were divided by the injection volume and multiplied by the simulation time step, ending up in an ash concentration in kg/mˆ3. Since this process wasn't described in the manuscript, we will add a short explanation in order to avoid confusion. Since the injection volume was approximated better or worse by the adaptive mesh dependent on the refinement level, it is true that the total injected amount of ash slightly differs for the different grid resolutions. But these differences are quite small. In the grid configuration used for most of the simulations (an adaptive mesh with a fine grid level of 17 and a coarse grid level of 8) as well as for a uniform mesh with a refinement level of 17, about 99.5 % of the calculated mass was included in the initial cloud area, while the uniform grid with a refinement level of 14 still comprised 97.6 % of the original mass.

2. In the abstract (page 1, line 3) and conclusions (page 18, line 5) the authors state that adaptive meshes are useful to resolve filament structures of volcanic emissions. However, in the chosen example and in the presented results no filaments are present. The authors need to better justify and motivate the selection of the Pinatubo eruption as a case study.

The reviewer is right in observing no fine filamentation in the experiment data. The Mount Pinatubo case was selected due to available data sets both in wind fields and initial conditions (injection rates) and coverage data. The argument of filamentation was used due to earlier experience with similar but fictional simulations (e.g. Behrens et al. 2000).

3. The model description (chapter 2) does not describe the semi-Lagrangian transport model in any detail. The reference (Behrens, 1996) is given but it is not clear from the text that the transport model is described in there. More details at least about the model concept are needed with a proper reference to the Behrens (1996) paper for further details.

While the authors did not feel that gap in the description and decided to omit such technical details for better readability and brevity, we are happy to add some more detail on the implementation and algorithmic details in a revised version of the paper (see also our comments on the review of A. Folch).

4. Add in the last paragraph of page 4 that ash is treated as a passive tracer. Particle deposition (page 4, line 28) was not monitored. It is unclear why, when it would have been as easy to implement as suggested in the conclusions (page 18, line 14-15). Since deposited ash is the main source of information for historical eruptions it would have been good to test the resolution dependence of that deposition.

The treatment as a passive tracer will be added to the revised manuscript. It is certainly true that more model validation – especially the evaluation of particle deposition – could be done. It is, however, technically quite complicated because of the mesh changing over time, which is the reason for omitting it at this point. As already explained when replying to the review of A. Folch the main purpose of this paper is to focus on the adaptive mesh methodology applied to the simulation of ash advection and sedimentation and not on the exact reproduction of a specific volcanic eruption.

5. The mass eruption rate in table 1 has wrong units. I would assume it is kg per second instead.

Yes, thank you, this is a mistake. We changed it to kg/s.

6. In table 2 add 'refinement' to the word level to avoid confusion with vertical levels.

Done.

7. A cloud radius in degrees is an odd choice since this means an elliptical shape in physical space. This is inconsistent with the stated initial radial expansion of the plume on page 5. Discuss and clarify.

Since the simulation grid is in degree, we decided to define the initial radius in degrees as well; and with an initial radius of only three degree the shape only deviates from a perfect circle by a few kilometers, which is below the highest resolution of the model. But for very explosive volcanic eruptions with an initial radius exceeding a few degrees, or for studies with a very high vertical resolution, it should be considered to define the initial radius in kilometers.

8. I disagree that results are converged for fine mesh levels larger than 16 (page 9, line1-2). According to figure 3, ash concentrations in the centre of the plume to the south west of the volcano increase significantly from fine mesh levels 17 to 20 and to 23.Quantify the differences and discuss convergence or non-convergence in greater detail.

The problem with convergence is that it can only be tested for smooth and therefore idealized data. So, we decided to address the convergence issue not in detail but to focus on qualitatively similar results at different refinement levels. What is meant is the fact that from refinement level 17 onward, qualitative differences between the levels are very minor. Based on this observation we decided to run most of the experiments on such this refinement level.

Other minor issues: We corrected the minor issues directly in the manuscript. A revised version will be uploaded following the Editors decision.

Page 1, line 17: fall out of tephra

Done!

Page 1, line 18: tephra fall(s) out can lead...

Done!

Page 2, line 2-3: add: ...warmer winters and colder summer on the Northern hemisphere continents through dynamical feedbacks and radiative forcing, respectively (Robock, 2000).

Done!

Page 2, line 4-6: timescales of minutes don't influence the diurnal cycle

It is written that tephra is remaining in the atmosphere on timescales of minutes to weeks, which influences the diurnal cycle.

Page 5, line 3: remove 'one of' since it has been said before that Pinatubo was the largest eruption in the 20th century.

No, it was not stated before, that the Pinatubo eruption was the largest during the 20th century. In fact according to the Smithsonian Catalogue the largest eruption of the 20th century was the Novarupta eruption of 1912, the Pinatubo eruption was the largest eruption in terms of stratospheric disturbances.

Page 6, line 5: what is 'atypical' about the winds?

Not corresponding to the prevailing southwest wind owing to the passing typhoon (compare Wolfe and Hoblitt, 1996).

Page 7, line 20: say already here that 7 times means refinement level 8 (in table 2).Write 'in the initial model domain before refinement'.

Will be corrected in the revised version of the manuscript.

Page 10, figure 3: increase font of colour bar and text.

Will be corrected in the revised version of the manuscript.

Page 11, figure 4: use identical and more meaningful colour bar. There are no yellow or red colours visible. What defines the surface of the ash cloud? If it is a threshold concentration the figure should show an iso-surface. Explain.

[Figure]

This and following comments refer to the colorbars of the figures. This was also noted by the other reviewer A. Folch. We will update the figures in the revised version of the manuscript.

Page 12, figure 5: use same colour bar for both panels to enable comparison.

See above, last comment.

Page 12, line 15: delete 'however'.

Done!

Page 12, line 11-17: use information from table 3 for superposition of different ash sizes. Ideally, this should give the best fit and allow for a more independent validation.

The authors are not sure what the reviewer suggests: Should we use all different particle sizes simultaneously and superposition the different results for the different percentages of their respective contribution? What the authors intended to express was that with a sinking rate corresponding to the mentioned particle size the best correspondence to observations could be observed.

Page 13, figure 6: use identical and more meaningful colour bars. Yellow and red colours not visible.

See comment above.

Page 15, line 16: 'since none of our (model) simulations'

Done!

Page 16, line 3-4: write: will (not might) be underestimated

Done!

Page 17, line 5-7 and page 18, figure 11: it is not obvious to me that the shape is recovered well in all calculations. Quantify differences, in particular, discuss differences between top right and bottom right panels (identical fine resolution). Label for bottom

right panel: shouldn't it read 'coarse=8'?

Will be updated in the revised manuscript.

Page 17, line 8-12: this is the common way to calculate performance gains due to adaptation. However, a transport model written and optimized for constant resolution can be significantly faster than an adaptive grid code run at constant grid resolution. Discuss to which extend this issue might apply here.

We assume that the performance of the semi-Lagrangian method employed here is relatively independent of the mesh design. Since we use a specialized algorithmic design that is based on a gather-scatter mechanism (see Behrens et al. 2005) for the ability to perform numerical operations on stride-one-vectors rather than unstructured meshes, earlier experiments have shown that the overhead imposed due to the adaptive mesh refinement is below some 5 % of the total run-time.

Page 17, line 12: unresolved reference

The following reference will be visible in a revised version: Madankan, R., Pouget, S., Singla, P., Bursik, M., Dehn, J., Jones, M., Patra, A., Pavolonis, M., Pitman, E., Singh, T., and Webley, P.: Computation of probabilistic hazard maps and source parameter estimation for volcanic ash transport and dispersion, Journal of Computational Physics, 271, 39 – 59, doi:http://dx.doi.org/10.1016/j.jcp.2013.11.032, http://www.sciencedirect.com/science/article/pii/ S0021999113007948, frontiers in Computational PhysicsModeling the Earth System, 2014.

References J. Behrens, N. Rakowsky, W. Hiller, D. Handorf, M. Läuter, J. Päpke, K. Dethloff (2005). amatos: Parallel Adaptive Mesh Generator for Atmospheric and Oceanic Simulation, Ocean Modelling, 12(1-2):171-183. J. Behrens, K. Dethloff, W. Hiller, A. Rinke (2000). Evolution of Small-Scale Filaments in an Adaptive Advection Model for Idealized Tracer Transport. Mon. Wea. Rev., 128:2976-2982. Wolfe, E. W. and Hoblitt, R. P (1996). Overview of the Eruptions. Quezon City : Philippine

Institute of Volcanology and Seismology ; Seattle : University of Washington Press, http://pubs.usgs.gov/pinatubo/wolfe/index.html.

We would like to thank M. Herzog for his thorough reading of the manuscript and his helpful and insightful comments and suggestions which will improve the manuscript if the editor requests a revised version.

---

## Author Response (AR1)

Dear Prof. Macedonio,

We have uploaded the revised version of the manuscript. Our response to the comments by A. Folch and M. Herzog as well as a marked-up manuscript are attached to our cover letter below. We have tried to incorporate most of the comments and suggestions by both reviewers. Details are given below in the detailed response.

The most relevant changes made in the manuscript are:
- We revised some of the figures to better show the different ash isosurfaces and to use more consistent colorbars (see Fig. 2 - 6, 11)

- We significantly extended the model description part (p. 3 and 4)

Furthermore several minor changes have been made to the manuscript in order to increase precision and clarity.

We would like to thank you, both reviewers as well as all persons involved in processing this manuscript and we honestly hope that you find our revisions of the manuscript sufficient and look forward to hear from you.

Kind regards

Elena Gerwing

**Authors reply to comments by A. Folch[1]**

1. *The model does not take into account for processes known to be relevant in ash cloud dynamics such as diffusion, dry/wet deposition mechanisms or ash aggregation. As a result, the model cloud dynamics limits to wind advection and particle sedimentation. Near-source effects (e.g. plume dynamics or gravity current) are ignored. On the other hand and more important, I do not understand why particle ground deposition is not contemplated. A part from the obvious interest of estimating ash/tephra fallout, the computation of the deposit is necessary (together with satellite imagery) for model validation (see below)*

   Indeed the model is simplified with respect to the ash cloud chemistry and microphysical processes. Our focus in this paper lies exclusively on the adaptive mesh methodology together with the simulation of sedimentation in context of the Lagrangian approach. Therefore processes like wet/dry deposition and ash aggregation are not considered. In order to consider plume dynamics and gravity currents other models like Atham or PDAC are certainly more appropriate to do so. After all, this is only an advective model like the basic Fall3D but with the add on of an adaptive mesh. Therefore we only focused on gravitational sedimentation further away from the plume stem.

   We have added one sentence into the introduction that we neglect this process as this is not the focus of the paper (p. 2, lines 31-33)

   The quantification of sedimentation to the ground (fall-out) could certainly be included but is at the moment not implemented in the model.

   Long term goal of our work is to include adaptive meshing into models like ATHAM or PDAC or any general GCM model. We consider this study as a first feasibility study for this endeavor and hope to encourage more scientists to work along these lines.

   We have added an extra paragraph at the end of the manuscript describing our roadmap for further developments (p. 21, lines 6-11)

2. *The innovative aspect of this manuscript is the use of adaptive grids in ash cloud simulations. However, several aspects regarding the numerical algorithm and its advantages are not detailed. Is the mesher embedded in the code? How often is the refinement applied? At each time integration step or at user-defined time intervals? The overhead of performing variable interpolation after each refinement and weather this is done in a conservative way or not is not mentioned.*

   You are right, we omitted these important aspects. But we will add this information in a revised version.
   The model uses a quasi-conservative semi-Lagrangian approach, as documented in (Behrens, 2006). This is a low order upstream interpolation method that can handle complex flow fields in 3D yet is simple enough not to impose too strict computational demands. The mesh refinement is triggered, as defined by the refinement criterion. As indicated in the manuscript, the tolerance for refinement is set to 0.02, which means that a grid cell is marked for refinement if the ash concentration gradient of the cell is above 2 percent of the maximum of all concentration gradients. The refinement criterion is calculated during each time step, but the actual refinement takes place only, if sufficiently many cells are marked for refinement. In our model at least 0.1 percent of the cells need to be marked for refinement to perform the refinement of the grid. The semi-Lagrangian scheme requires an interpolation in each step.

   The section describing the model has been significantly extended to incorporate the questions addressed by A. Folch (p. 3 and 4)

3. *The authors conclude that (pg 18, lines 1-4): "we have demonstrated the versatility of adaptive meshing algorithms for modeling the dispersion of volcanic emissions. Especially the high performance*
* * *
[1] Note that all references with pages and line numbers refer to the annotated manuscript.

*of this code would allow, if implemented into operational ash dispersion models, a significant improvement of dispersion predictions as model runs could be carried out significantly faster compared to codes using a fixed grid". Certainly there is a potential, but I think that this conclusion is too precipitated for many reasons. Constraining to Eulerian models (where this assertion would make sense), an important point not mentioned is code parallelism (most operational models actually run in parallel). The drawbacks of mesh adaptivity in the parallel execution of transient (time-evolving) problems are well-known: redefine the optimal domain decomposition at each refinement to ensure processor load balance has large associated interpolation and communication costs, code scalability breaks, etc. As a result, it is unclear whether the strategy would suppose a gain or not when running on hundreds of processors. This affects the main conclusion of this manuscript.*

While it is true that our simulations were carried out in serial mode, the code has also shown parallel efficiency (Behrens et al., 2005) and recently even more effective ways of parallelization have been presented (Behrens and Bader, 2009), all compatible with our algorithm. So, the authors do not see a general problem with parallelizing the code efficiently. The main message here is that even without a large parallel infrastructure, one can perform reasonably highly resolved simulations with just a laptop. We briefly mentioned in the manuscript the possibility of using the code in parallel.

Nothing has been changed in the manuscript to address this issue. We note that even in the org. manuscript we mentioned that the code could be executed in parallel (p. 2, line 22).

4. *Another controversial aspect which is not discussed by the authors is that "we apply the model for individual particle diameters and then combine the results of different runs to predict the sedimentation of the complete grain size distribution", pg. 4, line 22). I understand that this is necessary because the different particle sizes require of different refinements. However, how this affects efficiency in case on several (e.g. 10) particle classes is not even mentioned. Is the comparison shown in Figure 12 for a single class? This may not be fair, but it is difficult for a reader to extract conclusions since no details are given on the fraction of computation time of remeshing/interpolation. I suggest comparing the time of 10 simulations (coarse 8, fine 17) with that of a single run with all 10 classes and fixed grid.*

Currently the sedimentation of the particles is calculated by adding a vertical wind component to the wind field that is equivalent to settling velocity of that respective particle size. In principle the code could be rewritten to include an array of tracers of different size, each of which is advected with its individual wind field. The refinement criterion would then be applied to all tracer concentrations via the maximum gradient considering all tracers such that all tracers are well resolved.

We have added a comment on this in the conclusions (p. 20, lines 10 - 13).

5. *I missed details on the model numerical algorithm. Is it explicit or implicit? What about the time integration step (only 10 min is mentioned in pg. 4 line 17, but based on what?).*

The authors do not understand this remark. We use a semi-Lagrangian scheme. For the advection equation, this states that the upstream value of a material density needs to be preserved along trajectories. This in an unconditionally stable scheme.
Now, the conservative scheme does not only consider material particles, but cells, which may be distorted due to shear, convergence, divergence in the flow field. This imposes a stability restriction, since the cells are not allowed to degenerate in one time step. But under mild conditions on the regularity of the flow field this still gives a stable scheme. Therefore the time step is relatively irrelevant and given here for information only.

See the extended section on the model (p. 3 and 4).

6. *Model validation could certainly be improved. I wonder why the Pinatubo eruption was selected to this purpose given some obvious difficulties: the role of the gravity current (see Costa et al., Geophysical Research Letters, vol. 40, 1–5, doi:10.1002/grl.50942, 2013), the particular meteorological conditions at that time, the lack of extensive deposit sampling and inferred TGSD, etc. In any case, some quantitative model validation would be worth. On the other hand, it is stated (e.g. Table 4) that the refinement level goes down to <5 km. However, it seems that the driving meteorology is at 55 km and the wind field is linearly interpolated. Does it make sense? My impression is that refinement at sharp concentration gradients helps because it reduces numerical diffusion...*

We accept the criticism that more model validation could be done. However, since this study is mainly of methodological character and since we used a very simplified chemistry, we decided to focus on a more qualitative approach.
Regarding to the mesh refinement beyond the given wind field. This has been addressed in a former study (Behrens et al (2000)), were we showed that even in a very smooth/homogeneous flow field high resolution may be beneficial if shear in the flow field leads to stirring. So, we argue that even if the wind field does not contain small scale features, the transport benefits from high resolution. And yes, one of the effects that helps in this situation is the reduced numerical diffusion from high resolution.

Nothing has been changed in the manuscript to address this issue.

7. *Sensitivity study (section 4.1). The effect of variation in initial cloud height is actually a combined effect of injection height and driving meteorology (REMOTE)...*

We do not really understand, what the reviewer wants to express with this remark. Of course is this a combination of injection height with the wind field, and the simulated dispersion is a result of the prevailing wind conditions in the prescribed vertical layers of the injection.

As we were not clear about this comment nothing has been changed in the manuscript.

**Specific comments**

We corrected the minor issues directly in the manuscript. A revised version will be uploaded following the Editors decision.

*Pg. 2, line 16. Parenthesis in reference*

Done!

*Pg 2, line 26. "very low computational cost". This is too vague and generic: : :Also, values of "seconds" is not what Fig. 12 shows.*

In Fig. 12 we actually compare one run with an adaptive mesh with calculations on a uniform mesh. Clearly the adaptive mesh with the same refinement level as the calculation on a uniform mesh is 10 times faster. Admittedly it still takes close to an hour so we rewrote the sentence accordingly.

Has been rewritten (p. 2, lines 26/27)

*Pg. 3, lines 8-9. The resolution of 0.5x0.5 is that of REMOTE? If so, I do not understand which is the gain with respect to driving global ECMWF (Era-Interim?) data, already available at this resolution. Do you mean that the mesoscale simulation is not used to increase the wind field resolution?*

We used  model results from a mesoscale model instead of meteorological analysis data, as they offer more flexibility for potential future applications, such as higher temporal resolution, e.g. in one hour intervals, increased spatial resolution, e.g. up to 10 km (Langmann et al., 2009), and model output for processes not yet included, such as rain rate per layer for wet deposition.

Nothing has been changed.

*Pg. 3, line 12. Parenthesis in reference*

Done!

*Pg. 3, line 12. Like?*

We did not really understand this remark.

*Pg. 3., line 15. Model ! Domain*

Has been changed.

*Pg. 3, equations (1) and (3). Even if only advection is considered, shouldn't these equations include the terms C(div_u) (where u = u_REMOTE+u_t)? Is the wind from REMOTE divergence free? Can the z-gradient of the terminal settling velocity be ignored?*

We are not sure what the reviewer is referring to. The advection equation is applied to the particle concentration (see eq. 1 and 3), i.e. the divergence of the particle concentration field needs to be known at this point not the divergence of the velocity field, as would be the case if we solve for the Navier Stokes equation that includes the divergence of the wind velocity field.

This section has been rewritten and we hope this clarifies the issue (p. 3).

*Pg. 4, line 26: "Since this work is a first case study of the modeling of sedimentation of ash particles on an adaptive mesh the impact of rain on the sedimentation and aggregation of ash particles is neglected." This is ok, but I have concerns about how aggregation could ever be incorporated in a future using this strategy. With sedimentation each particle class has to have its own mesh (different model runs) and aggregation requires concurrency.*

We are certainly aware of this problem and in this version of the code this is not possible. See also answer to remark 4 above.

We made a comment at the end of the conclusion on how to incorporate sedimentation of multiple particle classes in one single run.

*Pg. 5, lines 12-13. Are hours correct?*

Yes, three hours after the onset of the climactic eruption (at 13:40) the intensity began to decline at 16:40 and nine hours after the onset (at 22:40) the climactic eruption phase ended.

Nothing has been changed.

*Pg. 6, line 5: "This atypical wind in the lower and middle troposphere caused the wide distribution of tephra in nearly all directions around the volcano". Was this a meteo effect or because the radial gravity current?*

We assume that it is mainly a meteorological effect (compare Wolfe and Hoblitt, 1996).

Nothing has been changed.

*Pg. 8, Table 4. The horizontal/vertical element aspect ratio seems very large for tetrahedral elements (_ 100). Any hint on mesh element quality? In case of small angle, can this lead to oscillations/convergence problems?*

Indeed the vertical vs. horizontal spatial scales are very distinct and the mesh quality in this respect is low. However, in the presented model, the mesh is used for interpolation and for maintaining conservation properties. For this type of application the mesh quality is of minor importance. Additionally, the non-uniform/anisotropic mesh supports the anisotropy in atmospheric scales (vertical vs. horizontal velocity ratio for example).

Nothing has been changed.

*Pg. 9, line 15. "concentration on the surface". What does it mean? Which value? How is this defined?*

As indicated by the colorbar, the surface of the cloud is supposed to have an ash concentration of 1x10^(-3) kg/m^3. Of course this value is quite arbitrary, but we had to pick one value.

We added a describing sentence to each figure caption displaying the ash concentration on isosurfaces.

*Figure 7 (and others). Why so many contours of observations? Only the corresponding to the time should be shown for clarity.*

We will fix the coloring of the contours in a revised version of the manuscript.

We marked the relevant contour line in red, see Fig. 7.

**References cited above**

J. Behrens (2006). Adaptive Atmospheric Modeling - Key Techniques in Grid Generation, Data Structures, and Numerical Operations with Applications. Lecture Notes in Computational Science and Engineering, 54, Springer Verlag, Berlin, Heidelberg.

J. Behrens, N. Rakowsky, W. Hiller, D. Handorf, M. Läuter, J. Päpke, K. Dethloff (2005). amatos: Parallel Adaptive Mesh Generator for Atmospheric and Oceanic Simulation, Ocean Modelling, 12(1-2):171-183.

J. Behrens and M. Bader (2009): Efficiency Considerations in Triangular Adaptive Mesh Refinement, Phil. Trans. R. Soc. A, 367:4577-4589, DOI:10.1098/rsta.2009.0175.

J. Behrens, K. Dethloff, W. Hiller, A. Rinke (2000). Evolution of Small-Scale Filaments in an Adaptive Advection Model for Idealized Tracer Transport. Mon. Wea. Rev., 128:2976-2982.

Langmann, B., M. Hort and T. Hansteen, Meteorological influence on seasonal and diurnal variability of Nicaraguan volcanic emission dispersion: A numerical model study, J. Volc. Geotherm. Res. 182, 34–44, 2009

Wolfe, E. W. and Hoblitt, R. P.: Overview of the Eruptions, Quezon City : Philippine Institute of Volcanology and Seismology ; Seattle : University of Washington Press, http://pubs.usgs.gov/pinatubo/wolfe/index.html, 1996.

**Authors Reply to comments by M. Herzog**

1. *The biggest issue is the model initialization and forcing. It is said (page 7, line 6-8) that the initial concentration is derived by dividing the eruption rate by the injection volume. However, this results in a flux of ash mass per unit volume and time, not a concentration. This flux should have been maintained over the duration of the corresponding eruption phase. However, it seems that an initial perturbation/concentration was used instead. If this is true, then this is a major flaw of the paper. Only adding the ash emitted within one second underestimates the erupted ash mass by many orders of magnitude. Releasing all ash erupted during each phase (as listed in table 1) instantaneously at the beginning of an eruption phase is equally unrealistic. This issue needs to be addressed, simulations repeated if needed before the manuscript can be published.*

   *Also, the total amount of ash should not depend on the choice of the grid. If the forced volume is different, then the ash flux into that volume should be (slightly) different to compensate.*

   The emitted ash per eruption rate was neither released instantaneously at the beginning of the eruption nor was only the ash amount emitted during one second introduced per time step. The calculated mass eruption rates in kg/s (listed in Table 1 in the manuscript) were divided by the injection volume and multiplied by the simulation time step, ending up in an ash concentration in kg/m^3. Since this process wasn't described in the manuscript, we will add a short explanation in order to avoid confusion.

   We added a sentence in order to clarify the procedure (p. 8, lines 8 – 10).

   Since the injection volume was approximated better or worse by the adaptive mesh dependent on the refinement level, it is true that the total injected amount of ash slightly differs for the different grid resolutions. But these differences are quite small. In the grid configuration used for most of the simulations (an adaptive mesh with a fine grid level of 17 and a coarse grid level of 8) as well as for a uniform mesh with a refinement level of 17, about 99.5 % of the calculated mass was included in the initial cloud area, while the uniform grid with a refinement level of 14 still comprised 97.6 % of the original mass.

   Compare page 9, lines 16 – 18.

2. *In the abstract (page 1, line 3) and conclusions (page 18, line 5) the authors state that adaptive meshes are useful to resolve filament structures of volcanic emissions. However, in the chosen example and in the presented results no filaments are present. The authors need to better justify and motivate the selection of the Pinatubo eruption as a case study.*

   The reviewer is right in observing no fine filamentation in the experiment data. The Mount Pinatubo case was selected due to available data sets both in wind fields and initial conditions (injection rates) and coverage data. The argument of filamentation was used due to earlier experience with similar but fictional simulations (e.g. Behrens et al. 2000).

   Nothing has been changed.

3. *The model description (chapter 2) does not describe the semi-Lagrangian transport model in any detail. The reference (Behrens, 1996) is given but it is not clear from the text that the transport model is described in there. More details at least about the model concept are needed with a proper reference to the Behrens (1996) paper for further details.*

   While the authors did not feel that gap in the description and decided to omit such technical details for better readability and brevity, we are happy to add some more detail on the implementation and algorithmic details in a revised version of the paper (see also our comments on the review of A. Folch).

4. *Add in the last paragraph of page 4 that ash is treated as a passive tracer. Particle deposition (page 4, line 28) was not monitored. It is unclear why, when it would have been as easy to implement as suggested in the conclusions (page 18, line 14-15). Since deposited ash is the main source of information for historical eruptions it would have been good to test the resolution dependence of that deposition.*

The treatment as a passive tracer has be added to the revised manuscript. (p. 6 line 11)

It is certainly true that more model validation – especially the evaluation of particle deposition – could be done. It is, however, technically quite complicated because of the mesh changing over time, which is the reason for omitting it at this point. As already explained when replying to the review of A. Folch the main purpose of this paper is to focus on the adaptive mesh methodology applied to the simulation of ash advection and sedimentation and not on the exact reproduction of a specific volcanic eruption.

Nothing has been changed in this regard.

5. *The mass eruption rate in table 1 has wrong units. I would assume it is kg per second instead.*

We changed it to kg/s.

6. *In table 2 add 'refinement' to the word level to avoid confusion with vertical levels.*

Done.

7. *A cloud radius in degrees is an odd choice since this means an elliptical shape in physical space. This is inconsistent with the stated initial radial expansion of the plume on page 5. Discuss and clarify.*

Since the simulation grid is in degree, we decided to define the initial radius in degrees as well; and with an initial radius of only three degree the shape only deviates from a perfect circle by a few kilometers, which is below the highest resolution of the model.

But for very explosive volcanic eruptions with an initial radius exceeding a few degrees, or for studies with a very high vertical resolution, it should be considered to define the initial radius in kilometers.

Nothing has been changed in this regard.

8. *I disagree that results are converged for fine mesh levels larger than 16 (page 9, line1-2). According to figure 3, ash concentrations in the centre of the plume to the south west of the volcano increase significantly from fine mesh levels 17 to 20 and to 23.Quantify the differences and discuss convergence or non-convergence in greater detail.*

The problem with convergence is that it can only be tested for smooth and therefore idealized data. So, we decided to address the convergence issue not in detail but to focus on qualitatively similar results at different refinement levels. What is meant is the fact that from refinement level 17 onward, qualitative differences between the levels are very minor. Based on this observation we decided to run most of the experiments on this refinement level.

Nothing has been changed in this regard.

Other minor issues:

We corrected the minor issues directly in the manuscript. A revised version will be uploaded following the Editors decision.

*Page 1, line 17: fall out of tephra*

Done!

*Page 1, line 18: tephra fall(s) out can lead...*

Done!

*Page 2, line 2-3: add: ...warmer winters and colder summer on the Northern hemisphere continents through dynamical feedbacks and radiative forcing, respectively (Robock, 2000).*

Done!

*Page 2, line 4-6: timescales of minutes don't influence the diurnal cycle*

It is written that tephra is remaining in the atmosphere on timescales of minutes to weeks, which influences the diurnal cycle.

Nothing has been changed.

*Page 5, line 3: remove 'one of' since it has been said before that Pinatubo was the largest eruption in the 20th century.*

No, it was not stated before, that the Pinatubo eruption was the largest during the 20$^{th}$ century. In fact according to the Smithsonian Catalogue the largest eruption of the 20$^{th}$ century was the Novarupta eruption of 1912, the Pinatubo eruption was the largest eruption in terms of stratospheric disturbances.

Nothing has been changed.

*Page 6, line 5: what is 'atypical' about the winds?*

Not corresponding to the prevailing southwest wind owing to the passing typhoon (compare Wolfe and Hoblitt, 1996).

Added explanation in manuscript.

*Page 7, line 20: say already here that 7 times means refinement level 8 (in table 2).Write 'in the initial model domain before refinement'.*

Corrected.

*Page 10, figure 3: increase font of colour bar and text.*

Done.

*Page 11, figure 4: use identical and more meaningful colour bar. There are no yellow or red colours visible. What defines the surface of the ash cloud? If it is a threshold concentration the figure should show an iso-surface. Explain.*

This and following comments refer to the colorbars of the figures. This was also noted by the other reviewer A. Folch.

Figures have been updated.

*Page 12, figure 5: use same colour bar for both panels to enable comparison.*

See above, last comment.

*Page 12, line 15: delete 'however'.*

Done.

*Page 12, line 11-17: use information from table 3 for superposition of different ash sizes. Ideally, this should give the best fit and allow for a more independent validation.*

The authors are not sure what the reviewer suggests: Should we use all different particle sizes simultaneously and superposition the different results for the different percentages of their respective contribution?

Our aim was to show the influence of using different particle sizes on the reproduction of the climactic eruption cloud. Obviosly, the best correspondence with observations would be obtained by using all measured particle classes with their corresponding percentage.

Nothing has been changed.

*Page 13, figure 6: use identical and more meaningful colour bars. Yellow and red colours not visible.*

See comment above.

*Page 15, line 16: 'since none of our (model) simulations'*

Done!

*Page 16, line 3-4: write: will (not might) be underestimated*

Done!

*Page 17, line 5-7 and page 18, figure 11: it is not obvious to me that the shape is recovered well in all calculations. Quantify differences, in particular, discuss differences between top right and bottom right panels (identical fine resolution). Label for bottom right panel: shouldn't it read 'coarse=8'?*

Figure 11 and description has been revised.

*Page 17, line 8-12: this is the common way to calculate performance gains due to adaptation. However, a transport model written and optimized for constant resolution can be significantly faster than an adaptive grid code run at constant grid resolution. Discuss to which extend this issue might apply here.*

We assume that the performance of the semi-Lagrangian method employed here is relatively independent of the mesh design. Since we use a specialized algorithmic design that is based on a gather-scatter mechanism (see Behrens et al. 2005) for the ability to perform numerical operations on stride-one-vectors rather than unstructured meshes, earlier experiments have shown that the overhead imposed due to the adaptive mesh refinement is below some 5 % of the total run-time.

Nothing has been changed as we consider this too technical to be incorporated into the manuscript.

*Page 17, line 12: unresolved reference*

Has been fixed.

[revised manuscript text omitted]